# Towards Reliable Identification of Diffusion-based Image Manipulations

**Alex Costanzino**[*]
University of Bologna

**Woody Bayliss**
BBC R&D

**Juil Sock**
BBC R&D

**Marc Gorriz Blanch**
BBC R&D

**Danijela Horak**
BBC R&D

**Ivan Laptev**
MBZUAI

**Philip Torr**
University of Oxford

**Fabio Pizzati**
MBZUAI

## Abstract

Changing facial expressions, gestures, or background details may dramatically alter the meaning conveyed by an image. Notably, recent advances in diffusion models greatly improve the quality of image manipulation while also opening the door to misuse. Identifying changes made to authentic images, thus, becomes an important task, constantly challenged by new diffusion-based editing tools. To this end, we propose a novel approach for ReliAble iDentification of inpainted AReas (RADAR). RADAR builds on existing foundation models and combines features from different image modalities. It also incorporates an auxiliary contrastive loss that helps to isolate manipulated image patches. We demonstrate these techniques to significantly improve both the accuracy of our method and its generalisation to a large number of diffusion models. To support realistic evaluation, we further introduce BBC-PAIR, a new comprehensive benchmark, with images tampered by 28 diffusion models. Our experiments show that RADAR achieves excellent results, outperforming the state-of-the-art in detecting and localising image edits made by both seen and unseen diffusion models. Further information about our code, data and models, including separate licensing terms, will be publicly available at https://alex-costanzino.github.io/radar/.

## 1 Introduction

With the advent of diffusion models [21, 15, 69, 52] and associated user-friendly tools [38, 47, 81, 89], image editing has never been so easy and powerful. Along with an unprecedented creative potential, such capabilities also imply risks of misuse. Today, even users with minimal technical expertise can produce highly realistic image edits, raising concerns about the trustworthiness of visual media. In particular, *text-based inpainting methods* [7, 6] can seamlessly insert arbitrary objects into existing scenes. Such edits pose substantial societal threats: for instance, malicious actors could insert compromising elements into a real picture, creating the potential for misinformation, false evidence, or reputational harm [77, 19, 34]. Moreover, removal of certain image elements may aid propaganda or help hide evidence [10]. These safety issues are addressed by approaches for Image Forgery Detection and Localisation (IFDL) [32, 45, 94, 39]. The objective of such methods is to determine whether images have been tampered by editing techniques and to localise modified regions, allowing for the identification of synthetically generated elements.

Given the growing number of image manipulation tools, generalisation of IFDL to the large variety of diffusion models presents a significant challenge. Existing IFDL methods focus mostly on Photoshop-inspired image manipulation – such as copy-pasting visual content – and have limited capabilities

---

[*]This work was carried out while the author was on a research visit at the University of Oxford.

39th Conference on Neural Information Processing Systems (NeurIPS 2025).

to tackle diffusion-based tampering [39, 45, 32, 94]. We argue that a critical issue is increasingly being overlooked: as the number of available diffusion models for image editing continues to grow, generalisation *across different diffusion models* becomes fundamental. An IFDL model trained on tampered images produced by a single diffusion model may completely fail when encountering images generated by a different model [87]. Given the high realism achieved by diffusion models and the very low barrier to their use – contrary to tools such as Photoshop – we believe there is a need for dedicated solutions *that specifically address robust IFDL for modern diffusion-based tampering*.

Modern IFDL methods often rely on automatically generated tampered images for training [32, 39, 45]. To effectively detect manipulations at test time, it is important to maximise the coverage of image editing models used during training [75]. However, this strategy alone is insufficient as many models remain private, customised, or expensive for large-scale data generation. As a consequence, generalisation to unseen models must be achieved through ad-hoc architectural design choices.

To address the above challenge, we propose RADAR (ReliAble iDentification of inpainted AReas), an IFDL method designed to detect and localise diffusion-based image manipulations in real scenarios. The design of RADAR focuses on two main principles: **(1)** learning effectively from data generated by multiple open-source inpainters, to be robust to a large number of models commonly used for image editing, and **(2)** maximising generalisation to unseen inpainters through targeted architectural choices. For achieving these goals, we propose several contributions. First, we design RADAR's architecture leveraging pre-trained foundation models for feature extraction, capitalising on their generalisation capabilities [56]. Importantly, we enable the use of multiple foundational encoders, capturing both semantic and structural features, which are subsequently fused through a trainable Fusion Block based on cross-attention. This multi-encoder design allows for richer feature representations, improving generalisation and absolute performance. The fused features are then processed by trainable components to extract localisation maps. We then employ a large number of diffusion models during training, maximising the diversity of training data. We impose contrastive constraints on this data, explicitly aligning the feature spaces associated with different inpainters. This enhances RADAR's ability to exploit training data generated with multiple models, along with promoting generalisation.

Finally, to show the need for cross-inpainter IFDL in realistic deployment conditions, we evaluate RADAR and baselines on BBC-PAIR (BBC - Paired Authentic and Inpainted References), a comprehensive benchmark built with 28 different inpainters to reflect real-world inference behaviours. We include three evaluation scenarios in BBC-PAIR, designed to mimic inpainting practices used by users editing images, such as model customisation.

Our contributions can be summarised as follows:

- We develop RADAR, a new method for IFDL designed for cross-inpainter robustness, leveraging multiple foundation models for feature extraction, and transformer-specific contrastive constraints explicitly designed to exploit data generated by multiple inpainters, significantly improving generalisation;
- We propose a novel data generation pipeline and construct BBC-PAIR, a new benchmark for training and evaluating methods for inpainting-based IFDL. BBC-PAIR includes over 150,000 generated images, 28 distinct inpainters, and three evaluation scenarios;
- We demonstrate state-of-the-art performances on all three BBC-PAIR scenarios, against 7 recent baselines, for both detection and localisation of tampered areas.

## 2    Related work

**Identifying synthetic images.**    Many methods have been developed to identify synthetic images [20, 13, 57, 80, 28, 83, 91, 23] without abilities to localise tampered regions. Instead, IFDL methods identify both tampered images and localised regions generated by diffusion models. The problem is typically addressed as a data-driven task, although there also exist training-free approaches [92, 66]. Some rely on the RGB modality only [63], while many [33, 32, 94, 46, 78, 18] employ low-level multi-modal fused features from spatial and frequency domains, to raise sensitivity to splicing and copy-move across images. This requires training of ad-hoc encoders, limiting generalisation. SAFIRE [45] uses foundation models, but still focuses on a single modality. Others [39, 86, 85] propose language model-based approaches, limited in scale as they require human-verified labels. RADAR instead

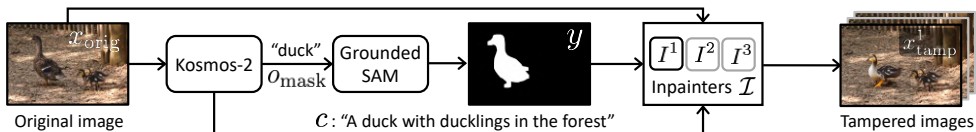

Figure 1: **Data generation pipeline.** We first extract with Kosmos-2 [58] a set of objects in an image $x_{\text{orig}}$, along with a caption $c$. We then use Grounded SAM [65] to segment one of the objects $o_{\text{mask}}$ (e.g. "duck"). Lastly, we use a set of $K$ text-to-image inpainters $\mathcal{I}$ to generate $K$ tampered images $\{x_{\text{tamp}}^k\}_{k=1}^K$. We use $c$ as a prompt to avoid the generation of objects irrelevant to the scene.

can be trained with automatically-generated data, and exploits multiple pre-trained foundational transformers, extracting variate sets of high-level features, ultimately benefiting performance.

**Generalisation in image forensics.** Diversity of data helps generalisation [71, 35], although the effects of distributional biases remain significant even for large-scale training [3]. This justifies the combination of large-scale training and generalisation techniques. In deepfake detection, this problem has been tackled, either by increasing the number of models used for data generation [25, 57] or with pipelines based on foundation models [54, 74]. In IDFL, instead, many approaches train on data generated by a restricted set of inpainters, limiting cross-model generalisability [32, 39, 45, 94, 18]. While some use contrastive learning for generalisation [45, 17], we exploit a novel transformer-specific contrastive strategy on patch representations, designed by reasoning about different distributions emerging while editing images with diffusion models.

**Evaluation of IFDL.** Benchmarks for image forensics have evolved together with the progress in tools for image manipulation. Initial efforts focused on splicing detection, with the Columbia [37], VIPP [8], and DSO-1 [12] datasets. The field then expanded with CASIA [24] and NIST16 [31], which incorporated diverse manipulation types including splicing, copy-move, and removal attacks, while COVERAGE [84] specifically targeted copy-move detection through semantically challenging examples. The advent of diffusion models made necessary a new generation of benchmarks [53, 32, 45, 39]: CocoGlide [32] introduced the first evaluation of diffusion-based inpainting artifacts, while only recently Safire-MS Expert [45] and SID-Set [39] addressed the broader challenges of detecting and localizing synthetic content across varying scales and model architectures. However, these benchmarks rely on a single inpainting model for image generation. Larger-scale efforts either focus on synthetic image detection only [36, 57, 87], or include a limited set of inpainters [18]. In contrast, our BBC-PAIR spans images modified by 28 inpainters from open source, LoRA-customised, and closed source, to effectively assess generalisation capabilities in realistic setups.

## 3  RADAR

We aim to identify a pixel-wise map of regions inpainted by a diffusion model on an input image $x$. We first generate training data suitable for the task with multiple inpainters, as shown in Figure 1 (Section 3.1). As illustrated in Figure 2, our model RADAR obtains patch features for input images using transformer-based foundation models (Section 3.2). This allows RADAR to benefit from the generalisation capabilities of foundation models, whilst capturing image tampering cues from multiple modalities. We fuse modality-specific representations with a custom attention-based Fusion Block (Section 3.2). In addition, to improve generalisation across inpainters, we exploit ad hoc contrastive constraints during training (Section 3.3). Finally, we produce a tampering map $\tilde{y}$ with a Localisation Head which is trained using our automatically generated dataset (Section 3.4). At inference, besides localising modified image regions $\tilde{y}$, we also aggregate pixel-wise predictions to compute a detection score, predicting whether the image has been tampered or not.

### 3.1  Data generation

Similar to other works [45, 32], we design an ad-hoc data generation pipeline, shown in Figure 1. We begin by sampling an image $x_{\text{orig}} \sim \mathcal{D}_{\text{orig}}$ from a dataset $\mathcal{D}_{\text{orig}}$, which contains only genuine images. The image $x_{\text{orig}}$ is processed with the Kosmos-2 [58] multi-modal language model, prompted to extract a list of object names present in the scene, denoted as $\mathcal{O}_x = \{o^1, o^2, \dots, o^n\}$, and a

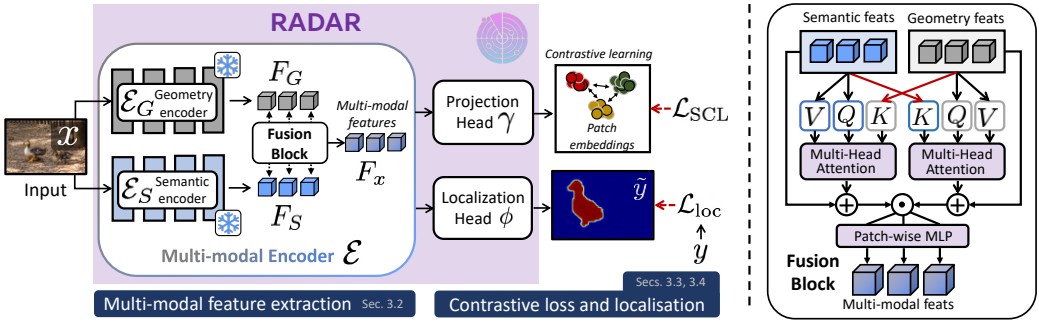

Figure 2: **Training RADAR.** We first extract, for an input image $x$, multi-modal features $F_x$ by using our multi-modal encoder $\mathcal{E}$. To implement $\mathcal{E}$, we employ the pre-trained Semantic Encoder $\mathcal{E}_S$ and Geometry Encoder $\mathcal{E}_G$, and fuse their extracted features with a Fusion Block (on the right, in details). The Fusion Block exploits a symmetric cross-attention mechanism, based on the swapping of keys $K$ in two multi-head attention mechanisms. For fusion, we use a patch-wise MLP. We process the resulting multi-modal features with a Localisation Head $\phi$, mapping to tampering maps $\tilde{y}$. Using $\tilde{y}$ and $y$, we back-propagate a localisation loss $\mathcal{L}_{\mathrm{loc}}$. To increase generalisation across inpainters, we use a Projection Head $\gamma$ to extract embeddings on which we impose a contrastive loss $\mathcal{L}_{\mathrm{SCL}}$.

caption $c$ describing the scene. For each image, we randomly sample an object name $o_{\mathrm{mask}} \sim \mathcal{O}_x$ (e.g. "duck") from the set. The selected object name $o_{\mathrm{mask}}$ is used as a prompt for Grounded SAM [65] (GS), which extracts a semantic mask of the object. We define a tampering mask $y$ as the mask extracted by GS, *i.e.* $\mathrm{GS}(x_{\mathrm{orig}}, o_{\mathrm{mask}})$. Then, we consider a set of $K$ text-based inpainters $\mathcal{I} = \{I^1, I^2, \dots, I^K\}$. For a given pair $(x_{\mathrm{orig}}, y)$, we apply each inpainter $I^k \in \mathcal{I}$ to obtain a tampered image $x_{\mathrm{tamp}}^k = I^k(x_{\mathrm{orig}}, y, c)$, where $c$ is the previously extracted caption, used as an inpainting prompt. This ensures that the inpainter generates plausible objects, increasing contextual coherence. For instance, in Figure 1, inpainting an aeroplane rather than a duck would impact realism. We also randomly replace $y$ with an algorithmically-generated mask to reduce semantic biases. More details in the Appendix.

For each original image $x \in \mathcal{D}_{\mathrm{orig}}$ we construct a sample $\mathcal{X} = \{x_{\mathrm{orig}}, x_{\mathrm{tamp}}^1, x_{\mathrm{tamp}}^2, \dots, x_{\mathrm{tamp}}^K, y\}$. By processing every $x \in \mathcal{D}_{\mathrm{orig}}$ in this way, we obtain a dataset $\mathcal{D} = \bigcup_{x \in \mathcal{D}_{\mathrm{orig}}} \mathcal{X}$ including original samples, the corresponding tampered images, and ground-truth masks.

## 3.2 Foundation models as feature extractors

**Multi-modal cues.** We build on the generalisation capabilities of foundational models [56] and extract image features using pre-trained visual encoders. Rather than using a single encoder, we fuse representations obtained from *two different models* focused on semantic and geometric image understanding. Manipulations may indeed exhibit semantic inconsistencies, such as incoherent textures, requiring adequate semantic-rich feature extraction for identification [49]. However, tampered regions may also reveal structural inconsistencies, such as violations of 3D coherence [70], making it identifiable only with a correct geometric interpretation of the scene. Importantly, these signals may not point to any single artefact, but emerge only when reasoning about the scene as a whole.

Following this, we define two pre-trained encoders for feature extraction. First, we employ a *Semantic Encoder $\mathcal{E}_S$*, *i.e.* a network trained to extract semantically rich features. Since self-supervised learning has been shown to promote the emergence of meaningful semantic representations [16], we adopt DINO-v2 [56] for $\mathcal{E}_S$. In parallel, to capture structural cues, we introduce a *Geometry Encoder $\mathcal{E}_G$*. Here, we use Depth Anything V2 encoder [88], which is well suited to extracting geometry-oriented features thanks to its large-scale pretraining for depth estimation. This choice of modalities is also inspired by relevant works [79, 29, 93]. Notably, both encoders share the same transformer backbone, making their embedding spaces naturally compatible for fusion, yet capturing different features.

In practice, we forward an input $x$ to both encoders, yielding $F_S = \mathcal{E}_S(x), F_G = \mathcal{E}_G(x)$. $F_S$ and $F_G$ refer to sequences of $N$ patch-based features extracted by the transformer encoders, so

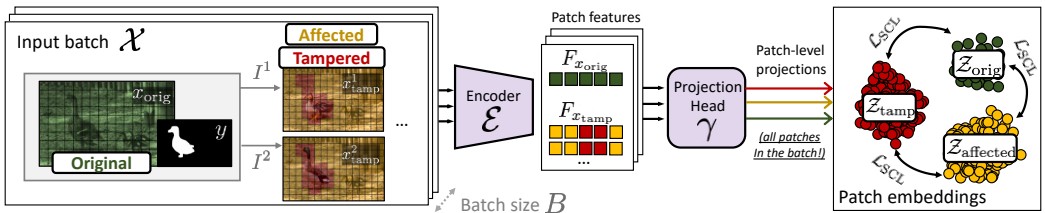

Figure 3: **Contrastive learning.** We identify **original**, **tampered** and **affected** patches. Corresponding feature patches are mapped by the head $\gamma$ to $\mathcal{Z}$ embeddings, on which we impose the contrastive loss $\mathcal{L}_{\text{SCL}}$, enforcing patch features of the same distribution to be clustered together. Note that we aggregate $\mathcal{Z}$ embeddings from multiple inpainters, promoting cross-inpainter generalisation.

$F_S = \{f_S\}_{n=1}^N$, $F_G = \{f_G\}_{n=1}^N$, with $f_G$ and $f_S$ being the individual patch-level features of each encoder, shown in Figure 2.

**Fusion block.** For multi-modal processing, we need to combine two sets of features. Inspired by previous works [76, 14], we fuse the patch-level features extracted by both encoders with a symmetric cross-attention mechanism. Intuitively, this compensates for the lack of information in each modality by attending to relevant visual cues in the other modality. We implement this using Multi-Head Attention (MH) layers, where we combine the keys $K$ of one modality with the queries $Q$ and values $V$ of the other, as shown in Figure 2 (right):

$$F_{S \leftarrow G} = \text{MH}_{S \leftarrow G}\left(K(F_S), Q(F_G), V(F_G)\right) \quad F_{G \leftarrow S} = \text{MH}_{G \leftarrow S}\left(K(F_G), Q(F_S), V(F_S)\right). \quad (1)$$

The $K(\cdot), Q(\cdot)$ and $V(\cdot)$ operators are implemented as multi-layer perceptrons (MLPs), separately for each modality. Then, we use another MLP to fuse information about corresponding features at the patch level. The MLP processes each pair of patches separately, hence being a *patchwise MLP* (Figure 2). We share the MLP weights across all pairs. A patch feature $f_M$ in the sequence is:

$$f'_S = f_S + f_{S \leftarrow G}, \quad f'_G = f_G + f_{G \leftarrow S}, \quad f_M = \text{MLP}\left([f'_S, f'_G]\right), \quad (2)$$

where $[\cdot]$ denotes concatenation. We set up the MLP so that $f_M$ has the same dimensionality as $f_S$ and $f_G$. We obtain $F_x$ by aggregating processed patches, $F_x = \{f_M\}_{n=1}^N = \mathcal{E}(x)$, abstracting our multi-modal feature extraction as $\mathcal{E}$.

## 3.3 Patch-level contrastive learning

**Contrastive distributions.** As mentioned in Section 3.1, for each $x_{\text{orig}}$ we generate multiple $\{x_{\text{tamp}}^1, ..., x_{\text{tamp}}^K\}$, with different models. Training on data from multiple models helps generalising in deepfake detection [57], but we experienced that naively using more inpainters for a fine-grained task, such as forgery localisation, leads to suboptimal performance. To mitigate this, we use contrastive learning. Intuitively, we aim to map features extracted from patches tampered by different models to a unique **tampered** distribution, and maximise their separation from an **original** distribution of untampered patches. Doing so, we encourage RADAR to focus on common characteristics *across diffusion models*, helping to exploit variable data. Importantly, this also aids generalisation to unseen models.

During each training iteration we sample a batch of $B$ samples $\{\mathcal{X}^1, \mathcal{X}^2, \ldots, \mathcal{X}^B\}$, where $\mathcal{X}^b \sim \mathcal{D}$, $b \in [1, B]$. For each tampered $x_{\text{tamp}}^k$ in the batch, we extract the indices of **tampered** patches, *i.e.* those that overlap with the inpainting mask. Likewise, we extract all patch indices from the original image as a representation of a **original** distribution. In Figure 3, we show how we classify patches following this definition. For clarity, we define $\mathcal{P}(x)$ as the set of patch indices corresponding to the patch grid of image $x$. That is, $\mathcal{P}(x)$ returns all patches spatially tiled over $x$, patch indices in the batch are defined as:

$$\mathcal{P}_{\text{tamp}} = \bigcup_{b=1}^{B} \bigcup_{k=1}^{K} \{p \in \mathcal{P}(x_{\text{tamp}}^{b,k}) | \sum_{i \in p} y_i^b > 0\}, \quad \mathcal{P}_{\text{orig}} = \bigcup_{b=1}^{B} \{p \in \mathcal{P}(x_{\text{orig}}^b)\}, \quad (3)$$

where $i \in p$ denotes all pixel indices within patch $p$, and $y_i$ is the value of the mask at pixel $i$.

Empirically, we find it beneficial to model a third category of patches, which we refer to as **affected**. These patches originate from tampered images but *do not overlap* with the inpainting mask, as we show in Figure 3 in yellow. Despite not being directly manipulated, such patches exhibit distributional shifts due to the auto-encoding process in latent diffusion models [68], which constitute the vast majority of open models. Hence, we define:

$$\mathcal{P}_{\text{affected}} = \bigcup_{b=1}^{B} \bigcup_{k=1}^{K} \{p \in \mathcal{P}(x_{\text{tamp}}^{b,k}) | \sum_{i \in p} y_i^b = 0\}. \tag{4}$$

**Contrastive constraint for transformers.** Next, we aim to separate distributions of the extracted patch features using a contrastive objective. We process the same batch of samples $\{\mathcal{X}^1, \ldots, \mathcal{X}^B\}$ following the multi-modal encoding described in Sec. 3.2, projecting the extracted patch-level features into a contrastive embedding space to enable distribution-level discrimination. For each image $x$ in the batch, where $x$ refers to either $x_{\text{orig}}$ or any tampered $x_{\text{tamp}}^k$, we compute its representation $F_x = \mathcal{E}(x) = \{f_M\}_{k=1}^{N}$. Each patch feature $f_M \in F_x$ is then forwarded to a Projection Head $\gamma$, implemented as a patch-wise MLP as before, yielding projected embeddings $Z_x$ that we can aggregate into a feature set $\mathcal{Z}$:

$$Z_x = \{\gamma(f_M)\}_{n=1}^{N} = \{z\}_{n=1}^{N}, \quad \mathcal{Z} = \bigcup_{b=1}^{B} \bigcup_{x \in \mathcal{X}^b} Z_x. \tag{5}$$

We write $\{z\}_{p \in \mathcal{P}}$ to denote the set of patch-level embeddings whose indices are in $\mathcal{P}$, and collect projected features associated with the patch index sets defined earlier:

$$\mathcal{Z}_{\text{orig}} = \{z\}_{p \in \mathcal{P}_{\text{orig}}}, \quad \mathcal{Z}_{\text{tamp}} = \{z\}_{p \in \mathcal{P}_{\text{tamp}}}, \quad \mathcal{Z}_{\text{affected}} = \{z\}_{p \in \mathcal{P}_{\text{affected}}}. \tag{6}$$

Ideally, as depicted in Figure 3, we would like each set of features to be well clustered, maximising its distance from others. This will ease the identification of affected patches [42]. We then enforce a supervised contrastive loss [43], treating each group as a distinct class. Let $\ell_z \in \{\texttt{orig}, \texttt{tamp}, \texttt{affected}\}$ denote the label associated with embedding $z \in \mathcal{Z}$. For each anchor $z_i \in \mathcal{Z}$, we define its anchor group $\mathcal{A}_i$:

$$\mathcal{A}_i = \{z_j \in \mathcal{Z} \setminus \{z_i\} \mid \ell_j = \ell_i\}, \tag{7}$$

and minimize the contrastive objective $\mathcal{L}_{\text{SCL}}$:

$$\mathcal{L}_{\text{SCL}} = \sum_{z_i \in \mathcal{Z}} -\frac{1}{|\mathcal{A}_i|} \sum_{z_j \in \mathcal{A}_i} \log \frac{\exp\left(z_i^\top z_j\right)}{\sum_{z_k \in \mathcal{Z} \setminus \{z_i\}} \exp\left(z_i^\top z_k\right)}. \tag{8}$$

Importantly, our contrastive learning imposes separation or aggregation on patch representations, hence *naturally exploiting transformer-based encoding*.

## 3.4 Localising tampering

**Training.** We aim to estimate tampering maps from input images. For simplicity, let us consider a single training sample $\mathcal{X} = \{x_{\text{orig}}, x_{\text{tamp}}^1, \ldots, x_{\text{tamp}}^K, y\}$. For each image $x \in \mathcal{X}$, we process the extracted multi-modal features $F_x = \mathcal{E}(x)$ through a Localisation Head $\phi$, implemented as a convolution followed by a sigmoid activation, following common practices in feature probing [11, 56]. This yields a tampering score map $\tilde{y} = \phi(F_x)$, as shown in Figure 2. At training time, we use both $x_{\text{orig}}$ and all $x_{\text{tamp}}$, defining a supervision mask $y_{\text{train}}$ based on whether $x$ is tampered or not, hence $y_{\text{train}} = y$ if $x \neq x_{\text{orig}}$, and $y_{\text{train}} = \mathbf{0}$ otherwise. We then define the localisation loss as the sum of a binary cross-entropy loss and a dice loss [73]:

$$\mathcal{L}_{\text{loc}}(x) = \mathcal{L}_{\text{BCE}}(\tilde{y}, y_{\text{train}}(x)) + \mathcal{L}_{\text{dice}}(\tilde{y}, y_{\text{train}}(x)). \tag{9}$$

Assuming a batched training, we can now impose our full objective:

$$\mathcal{L} = \mathcal{L}_{\text{SCL}} + \sum_{b=1}^{B} \sum_{x \in \mathcal{X}^b} \mathcal{L}_{\text{loc}}(x, y_{\text{train}}(x)). \tag{10}$$

Table 1: **Quantitative comparison.** We compare the three different setups of BBC-PAIR, *always significantly outperforming all other baselines*. As expected, we perform best on the ID set. Performance is also robust to the LoRA set, showing generalisation to customised models. In the presence of completely unseen inpainters (OOD), RADAR is still robust enough to outperform all competitors.

| Method | BBC-PAIR-ID | | | | BBC-PAIR-LoRA | | | | BBC-PAIR-OOD | | | |
| | Det. | | Loc. | | Det. | | Loc. | | Det. | | Loc. | |
| | AUC↑ | Acc↑ | F1↑ | IoU↑ | AUC↑ | Acc↑ | F1↑ | IoU↑ | AUC↑ | Acc↑ | F1↑ | IoU↑ |
|---|---|---|---|---|---|---|---|---|---|---|---|---|
| HiFi-Net [33] | 0.625 | 0.509 | 0.750 | 0.376 | 0.650 | 0.503 | 0.732 | 0.365 | 0.574 | 0.525 | 0.689 | 0.382 |
| TruFor [32] | 0.548 | 0.538 | 0.739 | 0.434 | 0.519 | 0.529 | 0.765 | 0.443 | 0.610 | 0.580 | 0.752 | 0.437 |
| MIML [63] | 0.468 | 0.483 | 0.214 | 0.110 | 0.517 | 0.500 | 0.240 | 0.131 | 0.629 | 0.505 | 0.187 | 0.098 |
| AdaIFL [48] | 0.533 | 0.498 | 0.762 | 0.383 | 0.540 | 0.532 | 0.763 | 0.412 | 0.606 | 0.554 | **0.791** | 0.447 |
| Mesorch [94] | 0.581 | 0.560 | 0.744 | 0.402 | 0.560 | 0.519 | 0.758 | 0.398 | 0.667 | 0.610 | 0.790 | 0.441 |
| SAFIRE [45] | 0.594 | 0.531 | 0.668 | 0.375 | 0.515 | 0.515 | 0.673 | 0.395 | 0.593 | 0.507 | 0.716 | 0.439 |
| SIDA [39] | 0.598 | 0.598 | 0.764 | 0.536 | 0.554 | 0.554 | 0.680 | 0.508 | 0.679 | 0.679 | 0.721 | 0.425 |
| RADAR | **0.945** | **0.931** | **0.832** | **0.600** | **0.901** | **0.773** | **0.810** | **0.521** | **0.805** | **0.709** | 0.785 | **0.450** |

During training, we back-propagate the total loss $\mathcal{L}$ to optimise the Localisation Head $\phi$, the Projection Head $\gamma$, and the Fusion Block, while keeping the multi-modal encoders frozen to preserve the modality-specific features they encode. Notably, since our contrastive strategy operates at the patch level, a batch yields a large number of samples in each $\mathcal{Z}$. This naturally benefits contrastive learning, which is known to improve with larger sets of comparisons [43].

**Inference.** At inference time, we discard $\gamma$ and use the Localisation Head $\phi$ to obtain a pixel-wise tampering probability $\tilde{y}$, which we binarise to produce tampering maps. Akin to others [22, 90], we define the probability of an entire image to be tampered as the mean of the top 1% highest values within $\tilde{y}$. This focuses on the most prominent forgery-related features while reducing the influence of low-confidence regions. We binarise this value with a threshold 0.5 to perform detection of tampered images.

## 4 Experiments

### 4.1 Setup

**BBC-PAIR** We use Section 3.1 to create the BBC-PAIR benchmark, allowing us to evaluate RADAR and baselines in three different setups. For each setup, we also include the original images $x_{\text{orig}}$ to evaluate performance in the absence of tampering. Note that BBC-PAIR also includes existing benchmarks.

***BBC-PAIR-ID*** We consider 15,000/100 randomly sampled OpenImages-v7 [50] as $\mathcal{D}_{\text{orig}}$ for train/test, and we use the data generation pipeline in Section 3.1 with 10 inpainters in $\mathcal{I}$: Stable Diffusion (SD) versions 1.4/1.5/2.1 [68], SD 3/3.5 [26], SDXL [62], Kandinsky 2.2 [64] and 3.1 [5], FLUX.1 [9] schnell and dev. All models are latent [68] and use Diffusers [61] inpainting pipelines, with parameters reported in the Appendix. We generate 150,000/1,000 images for training/testing, processing each image in $\mathcal{D}_{\text{orig}}$ with all $I \in \mathcal{I}$. Training on this set, we will be in-distribution (ID) for these inpainters.

***BBC-PAIR-LoRA*** We edit the $x_{\text{orig}}$ in the test split of BBC-PAIR-ID with 10 LoRAs, applied to SD1.5 [68], SD3.5 [26], SDXL [62] and FLUX.1 [9], generating 500 images. We report the LoRAs in the Appendix. Since we apply LoRAs to the inpainters of the ID set, this BBC-PAIR-LoRA set allows us to measure the effects of the model customisation on RADAR.

***BBC-PAIR-OOD*** To evaluate cross-inpainter generalisation, we synthesise data using *completely unseen* inpainters, hence having maximum distribution shift w.r.t. BBC-PAIR-ID. First, we generate 750 tampered images with 8 commercial inpainters: ClipDrop [72], Dall-E 2 [55], Adobe Firefly [1], FLUX.1 Fill [pro] [27], Ideogram [40], LightX [4], Phot.AI [60], YouCam [59]. We use DIV2K [2] as $\mathcal{D}_{\text{orig}}$ to maximise differentiation with the ID set. OOD evaluation also implies testing in uncontrolled scenarios. So, we select the existing inpainting-based CocoGlide [32], SafireMS-Expert [45] and SID-Set [39] test sets, to evaluate robustness to inpainting strategies different from ours. The union

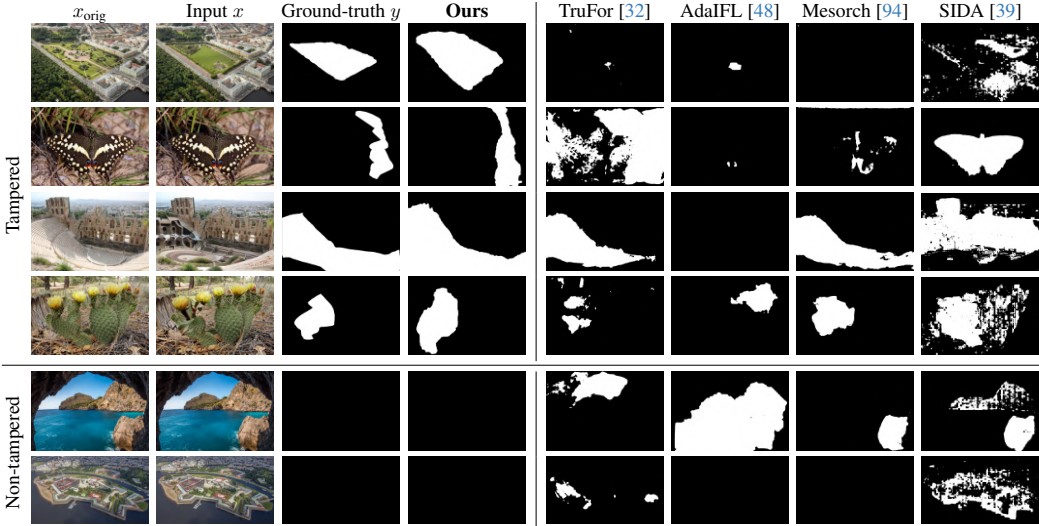

Figure 4: **Qualitative results.** Compared to the best baselines in the OOD scenario, RADAR accurately localises tampering. Baselines exhibit semantic biases, as SIDA (second row) and Mesorch (fifth row) identify full objects. Importantly, RADAR is robust to false positives and does not react to non-tampered images (last two rows). This explains our superior performance in detection.

of these samples is BBC-PAIR-OOD. *Detailed performance evaluation of all included datasets is in the Appendix.* We report only the average in the main paper due to space constraints.

**Baselines and metrics.** We exhaustively compare against the best recent open source methods: TruFor [32], HiFi-Net [33], SAFIRE [45], Mesorch [94], MIML [63], AdaIFL [48] and SIDA [39]. For localisation, we employ F1 and IoU to evaluate the similarity with the ground truth mask $y$, as commonly reported [39, 32, 45, 48]. Note that localisation is only evaluated on tampered images, akin to well-established practices [32, 39, 45, 94]. Consistent with recent studies [39, 32, 33, 94], we also evaluate on binary detection performances, using the area under the ROC curve (AUC) and accuracy with threshold 0.5 to evaluate the classification of images as tampered or not (Section 3.4). Since AdaIFL and MIML do not natively support forgery detection, we calculate a detection score as ours, selecting the most confident values of the tampering map $\tilde{y}$ (Section 3.4).

**Training details.** We train RADAR for 120 epochs with batch-size 16, dropout ($p = 0.1$) and NAdam optimiser [44] (lr=$10^{-4}$ and decay of $10^{-5}$) using 4 NVIDIA A100 80GB GPUs, optimizing the fusion block, $\gamma$ and $\phi$ while keeping $\mathcal{E}_S$ and $\mathcal{E}_G$ frozen. More details are given in the Appendix.

### 4.2 Comparisons with baselines

Table 1 reports experimental comparison with baselines. We *significantly outperform all methods* in both detection and localisation. As expected, we yield the highest metrics in the ID set. Note that while we have an advantage against baselines on ID data – since our training data are generated by the same inpainters – this allows us to be more robust in realistic scenarios. For the deployment of RADAR, *we aim to maximise performance on inpainters commonly used for image editing*, such as open source models. Since open source models allow data generation with ease, we argue that the best practice is to use them for training. Most importantly, RADAR also shines in generalisation, yielding the best localisation on both LoRA (**0.521** in IoU) and OOD sets (**0.450** in IoU).

Interestingly, in detection, all baselines suffer considerably, yielding low accuracies. We attribute this to their limited robustness to the distribution shift with respect to their training data. The better generalisation of RADAR, instead, is reflected in *highly superior detection performance*, improving AUC **+0.32**/**+0.25**/**+0.13** against the second best for ID/LoRA/OOD. We report detailed performance tables in the Appendix, including separate evaluations on each of the BBC-PAIR-OOD datasets. Since localisation performance is evaluated on tampered images only – following well-established practices [39, 32, 45, 94] – detection results *assess our robustness to false positives*. This is noticeable

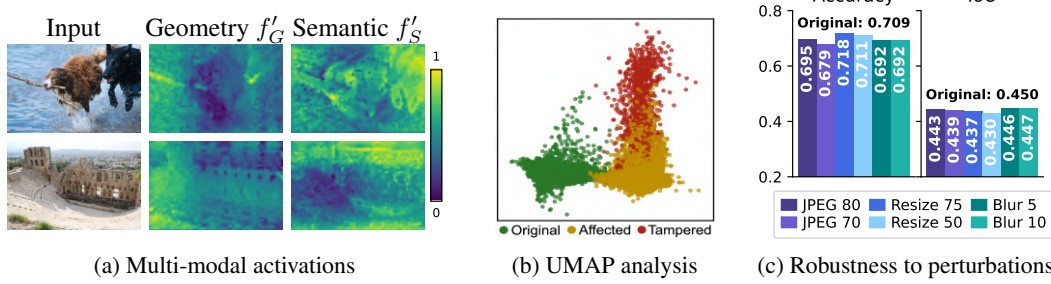

|                | Input | Geometry $f'_G$ | Semantic $f'_S$ |
| --- | --- | --- | --- |

(a) Multi-modal activations  (b) UMAP analysis  (c) Robustness to perturbations

Figure 5: **Properties of RADAR.** In Figure (a), we analyse feature magnitude in both encoders $\mathcal{E}_S$ and $\mathcal{E}_G$, highlighting that they are roughly complementary. In Figure (b), we show with UMAP [51] that **affected** patches serve as separators between **original** and **tampered**, justifying our boost in performance. In Figure (c), we demonstrate robustness to common web image operations, namely JPEG compression, resizing, and Gaussian blur.

in the qualitative results in Figure 4 on the BBC-PAIR-OOD set. Not only does RADAR produce more accurate masks to identify tampered regions, but it also correctly ignores non-tampered images (last two rows), where baselines produce predictions on objects in the scene instead (as the rock). SIDA also exhibits semantic biases (second row), identifying full objects as fake (the butterfly). This makes RADAR predictions more reliable for deployment, satisfying our initial design requirements. Note that we provide more qualitative and detailed comparisons in the Appendix.

### 4.3 Properties of RADAR

Next, we investigate some emerging properties of RADAR. For our investigation, we use the BBC-PAIR-OOD test set to minimise the distribution bias in the observed behaviour of the network.

**Multi-modality.** We display in Figure 5a the normalised activations of $f'_G$ and $f'_S$ in Equation (2) (hence, the features before fusion), averaged over channels for tampered inputs. Each map represents the importance of the features of each modality, per pixel, and ranges from 0 (low importance) to 1 (high importance). We notice that the spatial distribution of activations is roughly complementary across modalities. In other words, it shows that $\mathcal{E}_G$ and $\mathcal{E}_S$ tend to consider different elements for their feature extraction. This proves that our design choices, based on semantic and geometric encoders, allow us to extract meaningful features for the entirety of the scene, visually showing the reasons behind RADAR's performance. Interestingly, while semantic features focus on subjects (such as the dogs in the first row), geometric features highlight regular structures (such as the stairs in the second row). We conjecture that this may aid in detecting different types of tampering.

**Distribution analysis.** We want to understand the role of **affected** patches in the $\mathcal{L}_{\text{SCL}}$ optimisation. In Figure 5b, we report a UMAP [51] plot of the patch-level projection features extracted by $\gamma$ on BBC-PAIR-OOD. We observe that **affected** patches separate **original** and **tampered** clusters. This highlights that the hybrid nature of the **affected** patch distribution, processed by the auto-encoder in latent diffusion models, while containing no generated content, *naturally emerges during training* as an intermediate distribution, justifying our design.

**Robustness to perturbations.** In Figure 5c, we study RADAR performance in terms of Accuracy (detection) and IoU (localisation) on BBC-PAIR-OOD with common perturbations introduced by the upload on the web, such as JPEG compression (80/70 rate), resizing (75%/50%), or blurring with a Gaussian filter (variance 10/5). We highlight *negligible performance drops* in all tasks, further proving RADAR's robustness. We attribute this to the large pre-training of $\mathcal{E}_S$ and $\mathcal{E}_G$, presumably including perturbed images as well and consequently inducing robustness to perturbations in RADAR.

### 4.4 Ablation studies

We now present ablation studies on our contributions. Due to RADAR's training cost, for the following experiments, we train on $10\%$ of the original BBC-PAIR-ID for 70 epochs.

Table 2: **Ablation studies.** We investigate fusion strategies, contrastive formulations, and the number of inpainters ($K$). Our setup is always the best.

| Setup | BBC-PAIR-ID | | | | BBC-PAIR-LoRA | | | | BBC-PAIR-OOD | | | |
|---|---|---|---|---|---|---|---|---|---|---|---|---|
| | Det. | | Loc. | | Det. | | Loc. | | Det. | | Loc. | |
| | AUC↑ | Acc.↑ | F1↑ | IoU↑ | AUC↑ | Acc.↑ | F1↑ | IoU↑ | AUC↑ | Acc.↑ | F1↑ | IoU↑ |
| *Multi-modality and fusion* | | | | | | | | | | | | |
| $\mathcal{E}_S$ only | 0.582 | 0.502 | 0.780 | 0.538 | 0.590 | 0.510 | 0.735 | 0.484 | 0.567 | 0.516 | 0.706 | 0.410 |
| $\mathcal{E}_G$ only | 0.589 | 0.541 | 0.781 | 0.533 | 0.558 | 0.502 | 0.774 | 0.496 | 0.524 | 0.510 | 0.736 | 0.422 |
| Concat | 0.691 | 0.529 | 0.788 | 0.546 | 0.581 | 0.515 | 0.773 | **0.498** | 0.543 | 0.519 | 0.748 | **0.427** |
| Sum | 0.602 | 0.515 | 0.785 | 0.538 | 0.598 | 0.510 | 0.756 | 0.484 | 0.523 | 0.513 | 0.739 | **0.427** |
| **Ours** | **0.921** | **0.893** | **0.790** | **0.547** | **0.771** | **0.678** | **0.789** | 0.474 | **0.746** | **0.689** | **0.773** | 0.422 |
| *Contrastive Learning* | | | | | | | | | | | | |
| w/o $\mathcal{L}_{SCL}$ | 0.903 | 0.849 | 0.774 | 0.511 | 0.627 | 0.599 | 0.743 | 0.424 | 0.709 | 0.670 | 0.718 | 0.402 |
| w/o Affect. | **0.932** | 0.872 | **0.790** | **0.550** | 0.763 | **0.694** | 0.777 | 0.465 | 0.729 | 0.676 | 0.756 | 0.423 |
| Affect=Orig | 0.922 | 0.856 | 0.783 | 0.538 | 0.727 | 0.651 | 0.771 | 0.447 | 0.741 | 0.688 | 0.758 | **0.428** |
| **Ours** | 0.921 | **0.893** | **0.790** | 0.547 | **0.771** | 0.678 | **0.789** | **0.474** | **0.746** | **0.689** | **0.773** | 0.422 |
| *Number of inpainters K* | | | | | | | | | | | | |
| $K = 1$ | 0.896 | 0.793 | 0.804 | 0.485 | 0.692 | 0.582 | 0.766 | 0.406 | 0.720 | 0.623 | **0.809** | **0.425** |
| $K = 5$ | **0.937** | 0.858 | **0.811** | 0.524 | 0.763 | 0.625 | 0.774 | 0.423 | 0.741 | 0.636 | 0.794 | 0.421 |
| $K = 10$ (**Ours**) | 0.921 | **0.893** | 0.790 | **0.547** | **0.771** | **0.678** | **0.789** | **0.474** | **0.746** | **0.689** | 0.773 | 0.422 |

**Fusion Block.** We evaluate the architecture of our multimodal encoder in Table 2 (top). We first restrict the model to single encoders ($\mathcal{E}_S$ and $\mathcal{E}_G$ only), and observe *significant decrease in performance*, highlighting the importance of complementary encoders used by RADAR. This shows that both semantic and geometric cues are fundamental for a proper understanding of the scene in IFDL. We also investigate simpler fusion mechanisms, replacing the multi-head attention with simple Concat or Sum operations followed by an MLP. In all cases, performance drops significantly, especially for detection (AUC **0.921** vs. **0.691** and Acc. **0.893** vs. **0.541** of the best alternative). This shows that our Fusion Block is important for extracting meaningful cues from heterogeneous foundation models.

**Contrastive loss.** We investigate alternative setups of our contrastive formulation in Table 2 (middle). Removing $\mathcal{L}_{SCL}$ (w/o $\mathcal{L}_{SCL}$) yields the highest drop in performance (AUC **0.932** vs. **0.921** and Acc. **0.893** vs. **0.849** in detection, F1 **0.790** vs. **0.774** and IoU **0.550** vs. **0.511** in localisation), showing that only expanding $\mathcal{I}$ is suboptimal. We also investigate the effectiveness of the **affected** distribution, first removing it completely (w/o affected) and then mapping **affected** patches to the **original** class instead (Affect=Orig, see Section 3.3). All alternative configurations result in worse performance.

**Number of inpainters.** We test RADAR by reducing the number of inpainters $K$ in $\mathcal{I}$ used for training. We compare training with $K = \{1, 5, 10 \text{ (ours)}\}$ by randomly selecting $K$ inpainters over the full set of 10 inpainters used for the generation of BBC-PAIR-ID. We normalise the computational cost for training, and report the average over 3 runs in Table 2 (bottom). For both detection and localisation, increasing the number of inpainters *significantly boosts results* (Acc. **0.893** vs. **0.793** in detection, IoU **0.547** vs. **0.485** in localisation).

## 5 Conclusions

We introduced RADAR, a novel IFDL method designed to maximise in-distribution coverage and generalisation to unseen diffusion models. We proposed architectural contributions, fusing features extracted by multiple foundation models, and exploiting transformer-specific contrastive constraints on extracted patch features to promote generalisation. We also introduce BBC-PAIR, for a fair assessment in the presence of open-source models, model customisation, and commercial solutions. While we significantly outperform the state-of-the-art in IFDL, we emphasise that the complexity of the problem still leaves room for improvement. We hope that our research will encourage future work in this direction to mitigate the potential high societal impact of image manipulations.

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

# Appendix

In this document, we propose complementary information to the main paper. First, we list all assets used in Appendix A. Then, we report details for the reimplementation in Appendix B, including also details on the construction of BBC-PAIR. We provide extensive additional analysis, results, and ablations in Appendix C, while we propose final remarks and limitations in Section D.

## A  Assets

### A.1  Methods

For all the competitors [33, 32, 63, 48, 94, 45, 39], we employed their official implementations. In particular:

- Hifi-Net: `https://github.com/CHELSEA234/HiFi_IFDL` released under MIT license;
- TruFor: `https://github.com/grip-unina/TruFor` released under MIT license;
- MIML: `https://github.com/qcf-568/MIML` release under a custom licensing scheme;
- AdaIFL: `https://github.com/LMIAPC/AdaIFL` released under MIT license;
- Mesorch: `https://github.com/scu-zjz/Mesorch` released under MIT license;
- SAFIRE: `https://github.com/mjkwon2021/SAFIRE` released under CC BY-NC 4.0 license;
- SIDA: `https://github.com/hzlsaber/SIDA` released under a custom licensing scheme.

We thank the respective authors for making their code and pre-trained model weights publicly available, and for promptly and helpfully replying to our inquiries.

### A.2  Datasets

For the creation of BBC-PAIR, we gathered data from the following datasets:

- OpenImages-v7: `https://storage.googleapis.com/openimages/web/index.html`, released under Apache 2.0 license;
- CocoGlide: `https://github.com/grip-unina/TruFor` released under MIT license;
- SID-Set: `https://huggingface.co/datasets/saberzl/SID_Set` released under Creative Commons Attribution 4.0 International License;
- SafireMS-Expert: `https://www.kaggle.com/datasets/qsii24/safire-safirems-expert-multi-source-dataset` released under CC BY-NC 4.0 license.

Moreover, since SafireMS-Expert provided only tampered images, in order to be able to assess the detection performance between genuine and tampered images, we augmented the dataset with genuine images. These images come from DPREVIEW [30], the same source used to create the tampered ones, creating SafireMS-Expert++. We introduced a number of images equal to the number of tampered images present in SafireMS-Expert, obtaining a balanced set.

### A.3  Inpainters

To create BBC-PAIR, we employed open-source inpainters [68, 26, 62, 5, 64, 9], LoRA adaptations and commercial inpainters [72, 55, 1, 27, 40, 4, 60, 59]. In particular, for open-source inpainters, we leveraged:

- Stable Diffusion 1.4: `https://huggingface.co/CompVis/stable-diffusion-v1-4` released under CreativeML OpenRAIL-M license;
- Stable Diffusion 1.5: `https://huggingface.co/stable-diffusion-v1-5/stable-diffusion-v1-5` released under CreativeML OpenRAIL-M license;
- Stable Diffusion 2.1: `https://huggingface.co/stabilityai/stable-diffusion-2-1` released under CreativeML OpenRAIL-M license;
- Stable Diffusion 3: `https://github.com/replicate/cog-stable-diffusion-3` released under Apache 2.0 license;
- Stable Diffusion 3.5: `https://github.com/Stability-AI/sd3.5` released under MIT license;

- Stable Diffusion XL: `https://huggingface.co/docs/diffusers/en/using-diffusers/sdxl` released under MIT license;
- Kandinsky 2.2: `https://github.com/ai-forever/Kandinsky-2` released under Apache 2.0 license;
- Kandinsky 3.1: `https://github.com/ai-forever/Kandinsky-3` released under Apache 2.0 license;
- FLUX.1-schnell: `https://github.com/black-forest-labs/flux` released under Apache 2.0 license;
- FLUX.1-dev: `https://github.com/black-forest-labs/flux` released under non-commercial license;

Then, for LoRA adaptations, we employed:

- dAIversity Detailer 1.5: `https://www.shakker.ai/fil/modelinfo/3ecae938ab8c4aa4a65f6fc104aaad5f`;
- dAIversity SD3.5-Large-Photorealistic-LoRA: `https://huggingface.co/prithivMLmods/SD3.5-Large-Photorealistic-LoRA`;
- Flux-Detailer-LoRA: `https://huggingface.co/gokaygokay/Flux-Detailer-LoRA`;
- Dreamshaper SDXL-1-0: `https://huggingface.co/Lykon/dreamshaper-xl-1-0`;
- Juggernaut-XL-v6: `https://huggingface.co/RunDiffusion/Juggernaut-XL-v6`;
- Perfection 1.5: `https://civitai.com/models/411088?modelVersionId=486099`
- Perfection Flux: `https://civitai.com/models/411088?modelVersionId=931225`
- flux-RealismLora: `https://huggingface.co/XLabs-AI/flux-RealismLora`;
- ReaPhoLoRA 1.5: `https://civitai.com/models/59980/realistic-photography`
- Yamer's Realsim-v2 XL: `https://tensor.art/models/687689251510406883`;

Lastly, for commercial inpainters, we relied on:

- ClipDrop: `https://clipdrop.co/`;
- Dall-E 2: `https://openai.com/dall-e`;
- Adobe Firefly: `https://firefly.adobe.com/`;
- FLUX.1 Fill [pro]: `https://flux-ml.org/`;
- Ideogram: `https://ideogram.ai/`;
- LightX: `https://www.lightxapp.com/`;
- Phot.AI: `https://www.phot.ai/`;
- YouCam: `https://www.perfectcorp.com/consumer/apps/ymk`.

Each commercial inpainter has its own Terms of Service specified on their website.

# B   Details for reimplementation

## B.1   Construction of BBC-PAIR

Here, we provide detailed information on BBC-PAIR construction. The dataset generation process can be considered as two distinct tasks. The first task involves processing a predetermined cache of base images. These images can originate from any source. The second task involves taking the pre-processed base images and generating an inpainted counterpart for each image per inpainting model.

**Dataset pre-processing**   To begin, each image is checked for correct orientation, as inpainting models typically struggle with images that are not naturally oriented. This is achieved using the open-source library `https://github.com/ternaus/check_orientation`. Subsequently, each image is processed using Kosmos-2 [58], a model that extracts a wide range of information. For our purposes, we use the image descriptions and annotations provided by Kosmos-2. These annotations include the identification and localisation of objects within the image. Each object identified by Kosmos-2 is then evaluated as a candidate for inclusion in the dataset. The following steps are repeated for each such object:

1. **Segmentation**: the SAM [65] model is used to generate a segmentation mask for each object. It is prompted by Kosmos-2 using the object's location and description;

2. **Size filtering**: the area of the mask is calculated as a percentage of the image area. Objects with masks that are too small or too large are rejected. Specifically, any mask covering more than 83% or less than 0.23% of the image area is discarded. This filtering ensures that inpainted objects are neither minuscule nor encompass the entire image, a common issue with overgeneralised object descriptions from Kosmos-2;

3. **Mask cohesion**: once a mask of acceptable size is found, it is analysed for the number of disconnected components. This serves as a proxy for assessing the cohesiveness of the masked region. If a mask contains too many disconnected components, a dilation operation is applied—up to five times—until the number of components is reduced to fewer than eight. Every mask undergoes at least one dilation to improve cohesiveness for inpainting. If a mask passes all criteria, the following metadata is recorded:
   - Mask number;
   - Original mask;
   - Edited mask;
   - Masked object;
   - Masked area percentage;
   - Mask centroid coordinates.

In addition to object-based masks, a random mask is generated for each image; this is always the last mask generated. This is achieved by randomly selecting eight points and connecting them to form a polygon. The distribution of random mask areas across the dataset is matched to that of the object masks. Each image thus yields at least one random mask and up to ten object-based masks. This step also generates a master JSON file containing metadata for the entire dataset, which is required for the next phase.

**Dataset inpainting** The second phase of the process involves generating inpainted versions of the base images using one or more inpainting models listed in Appendix A.3. This step can be computationally expensive and is therefore designed to be easily distributed across multiple GPUs and compute nodes. Each inpainting job receives the master JSON file and a subset of the dataset to process. For each image, the model generates a new version using the image's full description and the associated mask, as specified in the JSON.

Due to computational constraints, only the first and last masks for each image are used during inpainting. This ensures that the random mask (always the last) is consistently processed, while the remaining object masks are retained for potential future use.

All inpainting models used are open source, run locally, and rely on the image's full description rather than object-specific text. Empirically, using the full image description yields better inpainting results.

To accommodate different model requirements, images are resized before processing. For models released prior to Stable Diffusion XL (SDXL), images are resized such that the longest edge is 512 pixels. Models released after SDXL use a resized length of 1024 pixels. This resizing enhances the inpainting performance of each model while maintaining consistency. Although inpainting at scale may introduce visual artefacts, the extensive size of the dataset is expected to statistically offset these anomalies.

Once all nodes complete processing, the inpainted images are incorporated into the master JSON file, and all data is aggregated.

## B.2 Additional training details

To implement our Multi-modal Encoder, we employ DINO-v2 ViT-B/14 [56] pre-trained on a large, curated and diverse dataset of 142 million images, comprising ImageNet-22k and Depth Anything V2 ViT-B/14 [88], pre-trained on a large mixed dataset comprising millions of real-world (*e.g.*, NYU Depth V2, KITTI, MegaDepth) and synthetic (*e.g.*, BlendedMVS, Virtual KITTI) depth images, enhanced with self-supervised learning on unlabeled web-scale data. The proposed Fusion Block employs bidirectional cross-attention between Semantic and Geometry features. Since the two embedding spaces yield features with different magnitudes, the feature embeddings are layer-normalised before being processed by the Multi-Head Attention heads to avoid overpowering one modality. The features are then processed via two Multi-Head cross-attention layers, with 8 heads

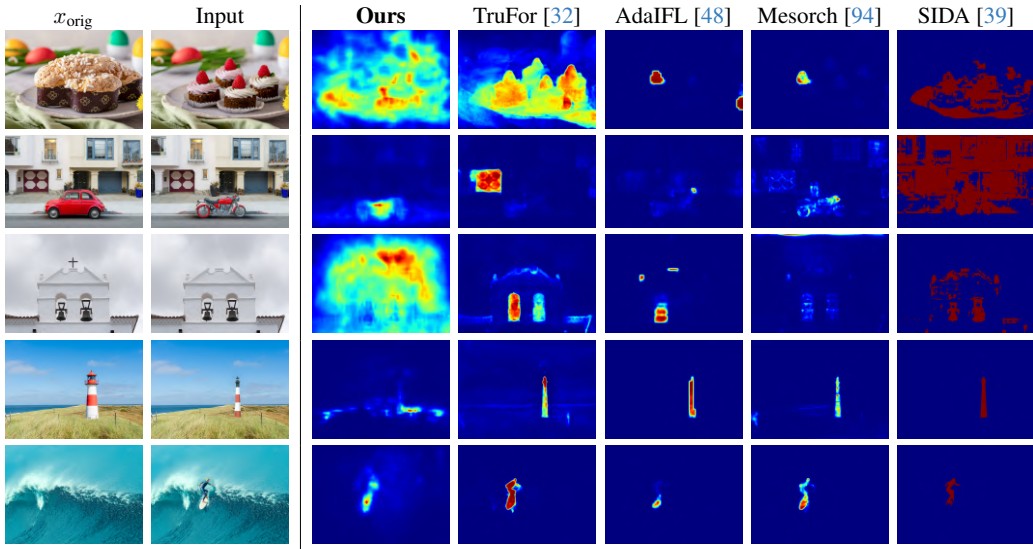

| $x_{\text{orig}}$ | Input | **Ours** | TruFor [32] | AdaIFL [48] | Mesorch [94] | SIDA [39] |

Figure 6: **Qualitative results on tampered in-the-wild images.** Note that SIDA produces a binary output.

each and residual connections, where each modality attends to the other. The mixed features are projected through a single-layer GELU-activated MLP and lastly layer-normalised.

**Machine configuration** Experiments were conducted on a high-performance computing server equipped with four NVIDIA A100 GPUs (80 GB each, PCIe) and a 96-core CPU, with a total of 866 GB of RAM. The system ran with NVIDIA Driver version 550.144.03 and CUDA Driver version 12.4.

## C  Additional analysis

### C.1  In-the-wild test

We report in Fig. 6 some qualitative samples on in-the-wild tampered images. To obtain these images, we hired a graphic designer with knowledge of inpainting technologies and asked them to download some images from the internet, and modify them with any tool of their choice, as long as it is generative inpainting. By doing so, we obtained 5 pairs of original/tampered images with human-in-the-loop tampering. Please note that we do not have ground truth tampering masks for this task, since many tools do not support saving them. We report original and tampered images, along with the probabilistic prediction localisation maps, in Figure 6. We notice once again how competitors tend to perform well in the presence of a strong semantic tampering (e.g., row 3 and row 4), while failing in the presence of less semantically biased tampering (e.g., row 1, row 2, and row 3). Conversely, RADAR produces a reasonable output in the presence of diverse manipulations, hence being more robust for in-the-wild deployment.

### C.2  Impact of the mask size

We propose an additional analysis on the impact of the mask size. We divide the available masks in BBC-PAIR into three sets – "small", "medium", "large" – by ranking them by number of occupied pixels and selecting the corresponding subset. We then evaluate RADAR on the three subsets, reporting results in Table 3. The results show that the detection performance is stable across all sizes. Regarding the localisation performance, on average, we perform better on small masks on BBC-PAIR-ID, while we perform better on large masks on BBC-PAIR-LoRA and BBC-PAIR-OOD. Since there is no clear trend across the three splits, we ascribe these differences to the nature of the specific tamperings rather than their sizes.

Table 3: **Impact of mask size at inference time.**

| Setup | ID | | | | LoRA | | | | OOD | | | |
|---|---|---|---|---|---|---|---|---|---|---|---|---|
| | AUC↑ | Acc↑ | F1↑ | IoU↑ | AUC↑ | Acc↑ | F1↑ | IoU↑ | AUC↑ | Acc↑ | F1↑ | IoU↑ |
| Small masks | 0.951 | 0.954 | 0.830 | 0.704 | 0.901 | 0.773 | 0.720 | 0.533 | 0.804 | 0.712 | 0.684 | 0.416 |
| Mid masks | 0.951 | 0.953 | 0.788 | 0.665 | 0.902 | 0.772 | 0.599 | 0.427 | 0.804 | 0.706 | 0.564 | 0.378 |
| Large masks | 0.950 | 0.954 | 0.828 | 0.551 | 0.902 | 0.773 | 0.864 | 0.539 | 0.805 | 0.709 | 0.834 | 0.456 |
| All masks | 0.945 | 0.931 | 0.832 | 0.600 | 0.901 | 0.773 | 0.810 | 0.521 | 0.805 | 0.709 | 0.785 | 0.450 |

Table 4: **RADAR's performance in low-resource environments.**

| Regions | AUC↑ | Acc↑ | F1↑ | IoU↑ |
|---|---|---|---|---|
| All | 0.932 | 0.821 | 0.802 | 0.545 |
| Single region | 0.930 | **0.829** | 0.795 | 0.540 |
| Multiple region | **0.960** | 0.712 | **0.898** | **0.622** |

## C.3 Impact of the number of forgeries

Since the Safire-MS Expert subset from MIBench-OOD specifically includes multi-source forgeries, we analysed RADAR's performance on Single Region forgeries (a single connected component in the ground truth) and Multiple Regions (more than one connected component in the ground truth), in Tab. 4. We observe that RADAR's performance remains competitive in both the presence of single regions and multiple regions inpainted. We hypothesise that this is due to the patch-level features separation promoted by our contrastive formulation.

## C.4 Performance on non-inpainting datasets

Although RADAR is designed for inpainting tampering only, and trained on inpainting data, we tested its performance on datasets including non-inpainting modifications, namely COVERAGE [84], CASIA-v2 [24], IMD2020 [53], CMFD [67], AutoSplice [41], and DSO [12]. We report results in Table 5. Our performance in localisation is not impacted, in which we achieve even higher F1 than in BBC-PAIR. This is due to the less refined copy-paste operations in these datasets, which, compared to inpainting ones, do not allow for a smooth blending of inserted elements, easing localisation. Importantly, this proves our point about building a dataset specifically focused on inpainting-based modifications. However, we also report lower detection performance. We believe this is due to the lack of distinction between genuine, affected and tampered pixels in non-inpainting modifications, causing outliers in detection. However, since the performance in localisation is satisfactory, a calibration of the detection score could suffice to improve the results.

Interestingly, baselines may report higher accuracies in splicing-based datasets, such as TruFor (*e.g.* AUC/accuracy on COVERAGE 0.770/0.680 in [32]), while falling short on inpainting-based benchmarks such as BBC-PAIR (see Tables 1). We attribute this to the different features important for detection, further motivating our choice to develop an inpainting-specific method for IFDL.

## C.5 Additional ablation studies

**Detection mechanism ablation** We report in Tab. 6 the results obtained by adding an additional classification head to our framework, trained with a binary cross-entropy with image-level labels. This is inspired by alternative frameworks employing similar methodologies [32]. We yield better detection results at the expense of worse localisation results. We observe then a trade-off between the detection and localisation performance in RADAR: in principle, one could train the framework either with a dedicated classifier or not, based on the performance that needs to be prioritised.

**Training with random and semantic masks** During training of RADAR, we use both semantic masks generated by Grounded SAM and random ones, as we describe in Section 3.1 of the main paper. We report in Tab. 7 the results of training RADAR either with semantic mask-only, random mask-only and mixing both random and semantic masks. We observe how the use of random masks, either

Table 5: **Performance of RADAR on non-inpainting datasets.**

| Dataset | AUC↑ | Acc↑ | F1↑ | IoU↑ |
|---|---|---|---|---|
| COVERAGE | 0.523 | 0.535 | 0.879 | 0.448 |
| CASIA-v2 | 0.505 | 0.505 | 0.903 | 0.452 |
| IMD2020 | 0.597 | 0.540 | 0.899 | 0.454 |
| CMFD | 0.517 | 0.512 | 0.944 | 0.474 |
| AutoSplice | 0.845 | 0.743 | 0.747 | 0.501 |
| DSO | 0.487 | 0.505 | 0.854 | 0.427 |

Table 6: **Ablation on the use of a dedicated classification head.**

| Setup | ID | | | | LoRA | | | | OOD | | | |
|---|---|---|---|---|---|---|---|---|---|---|---|---|
| | AUC↑ | Acc↑ | F1↑ | IoU↑ | AUC↑ | Acc↑ | F1↑ | IoU↑ | AUC↑ | Acc↑ | F1↑ | IoU↑ |
| Dedicated `cls` head | **0.957** | **0.898** | 0.785 | **0.547** | **0.810** | **0.681** | 0.753 | 0.431 | **0.769** | **0.695** | 0.745 | 0.417 |
| **Ours** | 0.921 | 0.893 | **0.790** | **0.547** | 0.771 | 0.678 | **0.789** | **0.474** | 0.746 | 0.689 | **0.773** | **0.422** |

on their own or mixed with semantic ones, improves the performance, especially in generalisation. Indeed, we argue that the use of random masks at training time aids the model in detaching the semantic bias interwoven with the task. In fact, while the forgery localisation is highly semantic, it is not guaranteed that the forgeries are produced with the semantics of the scene in mind, differentiating the task from semantic segmentation.

**Semantic encoder ablation**   In our implementation, we chose DINO-v2 primarily due to architectural and resolution compatibility with DepthAnything-v2, which facilitates a more intuitive fusion of features in our cross-attention mechanism, and favourable performance compared to CLIP on self-supervised learning benchmarks requiring semantic understanding [11]. However, CLIP's language supervision may offer rich semantics as well. We conducted an experiment using CLIP (ViT-B/16) instead of DINO-v2, and found that while performance remained reasonable, as shown in Tab. 8, it was slightly lower compared to DINO-v2 in our setup.   Importantly, the performance remained competitive with respect to the inference with DepthAnything-v2 only, proving the effectiveness of our novel fusion strategy. Nevertheless, CLIP remains a viable alternative, and we see potential in future exploration of multi-modal supervision within our framework.

## C.6   Features and attention analysis

We report in Fig. 7 more examples of the normalised activations of $f'_G$ and $f'_S$ (features before fusion), averaged over channels for tampered inputs. We once again highlight that the spatial distribution of activations is roughly complementary across modalities.

We also report the attention maps, $f_S \leftarrow f_G$ and $f_G \leftarrow f_S$, from the Fusion Block. These attention maps are obtained by averaging the values of the attention over the keys. In particular:

- $f_S \leftarrow f_G$, which is the mean of semantic values over geometry keys, represents which semantic features are actively using geometry features;
- $f_G \leftarrow f_S$, which is the mean of geometry values over semantic keys, represents which geometry features are actively using semantic features.

Also in this case, we observe that the spatial distribution of attention is roughly complementary. Moreover, despite the attention being mostly concentrated in the prediction area, we observe peaks also outside, highlighting that the framework is actively using information from the background as well for the prediction.

## C.7   Inference time and memory occupancy

We report in Tab. 9 the inference time per sample in `ms`, the memory occupancy of the forward pass in `GiB` and the input resolution in `px` for each considered method. We remind the reader that we test each method using the suggested resolution in the original papers. For each method, we compute the inference time, from when the sample is available on the GPU to the computation of the localisation

Table 7: **Ablation on the use of semantic and random masks at training time.**

| Setup | ID | | | | LoRA | | | | OOD | | | |
|---|---|---|---|---|---|---|---|---|---|---|---|---|
| | AUC↑ | Acc↑ | F1↑ | IoU↑ | AUC↑ | Acc↑ | F1↑ | IoU↑ | AUC↑ | Acc↑ | F1↑ | IoU↑ |
| Semantic masks-only | 0.965 | 0.884 | **0.805** | 0.519 | **0.791** | 0.609 | 0.764 | 0.405 | **0.785** | 0.656 | 0.796 | 0.417 |
| Random masks-only | **0.966** | **0.896** | 0.770 | 0.466 | 0.781 | 0.586 | 0.762 | 0.400 | 0.763 | 0.642 | **0.801** | **0.423** |
| **Ours** | 0.921 | 0.893 | 0.790 | **0.547** | 0.771 | **0.678** | **0.789** | **0.474** | 0.746 | **0.689** | 0.773 | 0.422 |

Table 8: **RADAR's performance with CLIP as Semantic Encoder.**

| Setup | ID | | | | LoRA | | | | OOD | | | |
|---|---|---|---|---|---|---|---|---|---|---|---|---|
| | AUC↑ | Acc↑ | F1↑ | IoU↑ | AUC↑ | Acc↑ | F1↑ | IoU↑ | AUC↑ | Acc↑ | F1↑ | IoU↑ |
| CLIP+DA-v2 | 0.901 | 0.844 | **0.802** | 0.503 | 0.691 | 0.596 | 0.758 | 0.402 | 0.706 | 0.627 | **0.799** | 0.412 |
| DINO-v2+DA-v2 (**Ours**) | **0.921** | **0.893** | 0.790 | **0.547** | **0.771** | **0.678** | **0.789** | **0.474** | **0.746** | **0.689** | 0.773 | **0.422** |

map, after a GPU warm-up, synchronising all threads before estimating the total inference time. While RADAR is slower than networks trained from scratch, other approaches based on foundation models [45] or LLMs [39] are considerably slower.

As an additional effort to further reduce computational cost, we propose an experiment substituting the DINO-v2 and DepthAnything-v2 backbones with their smaller variants, such as using ViT Small instead of ViT Base. We train this smaller model and compare it using the reduced training protocol described in Sec. 4.4 due to the training times of the full model. This substitution, whose results are depicted in, represents a good trade-off between performance and efficiency. In particular, this variant yields a runtime memory footprint of 3.106 GiB and an inference time of 99.424 ms, representing a 30% reduction in memory and a 57% reduction in inference time compared to the version presented in the main manuscript. Moreover, we are still very competitive in terms of accuracy and localisation compared to baselines in Table 1, even considering the reduced training setup.

In conclusion, RADAR performs competitively in low-resource environments.

### C.8 Additional qualitative results

We show in Fig. 8 more qualitative results coming from the commercial inpainters of the BBC-PAIR-OOD split of our benchmark. Moreover, we show in Fig. 9 qualitative results coming from the other datasets of the BBC-PAIR-OOD split, namely CocoGlide, SafireMS-Expert++ and SID-Set. Also, in Fig. 10 we show qualitative results coming from the BBC-PAIR-ID and BBC-PAIR-LoRA splits of our benchmark.

### C.9 Detailed quantitative results

We report in Tab. 11, Tab. 13, Tab. 12, Tab. 14 the full results of the evaluation performed in Tab. 1. In Tab. 14 we also show the official result reported from the competitors' papers, highlighting how they are close to our run. We also highlight that SafireMS-Expert++ is in-domain for SAFIRE, as well as SID-Set is in-domain for SIDA. Despite this advantage, we outperform them in the BBC-PAIR-OOD benchmark.

### C.10 Additional results on OpenSDI

We report in Tab. 15 additional results on the OpenSDI dataset [82]. Similarly to BBC-PAIR, OpenSDI contains images inpainted employing different diffusion models, but these are only a subset of the one proposed in BBC-PAIR (5 vs. 28), hence, having much less coverage.

## D Final remarks

### D.1 Limitations

While RADAR achieves state-of-the-art performance in image forgery detection and localisation, several limitations highlight avenues for future work:

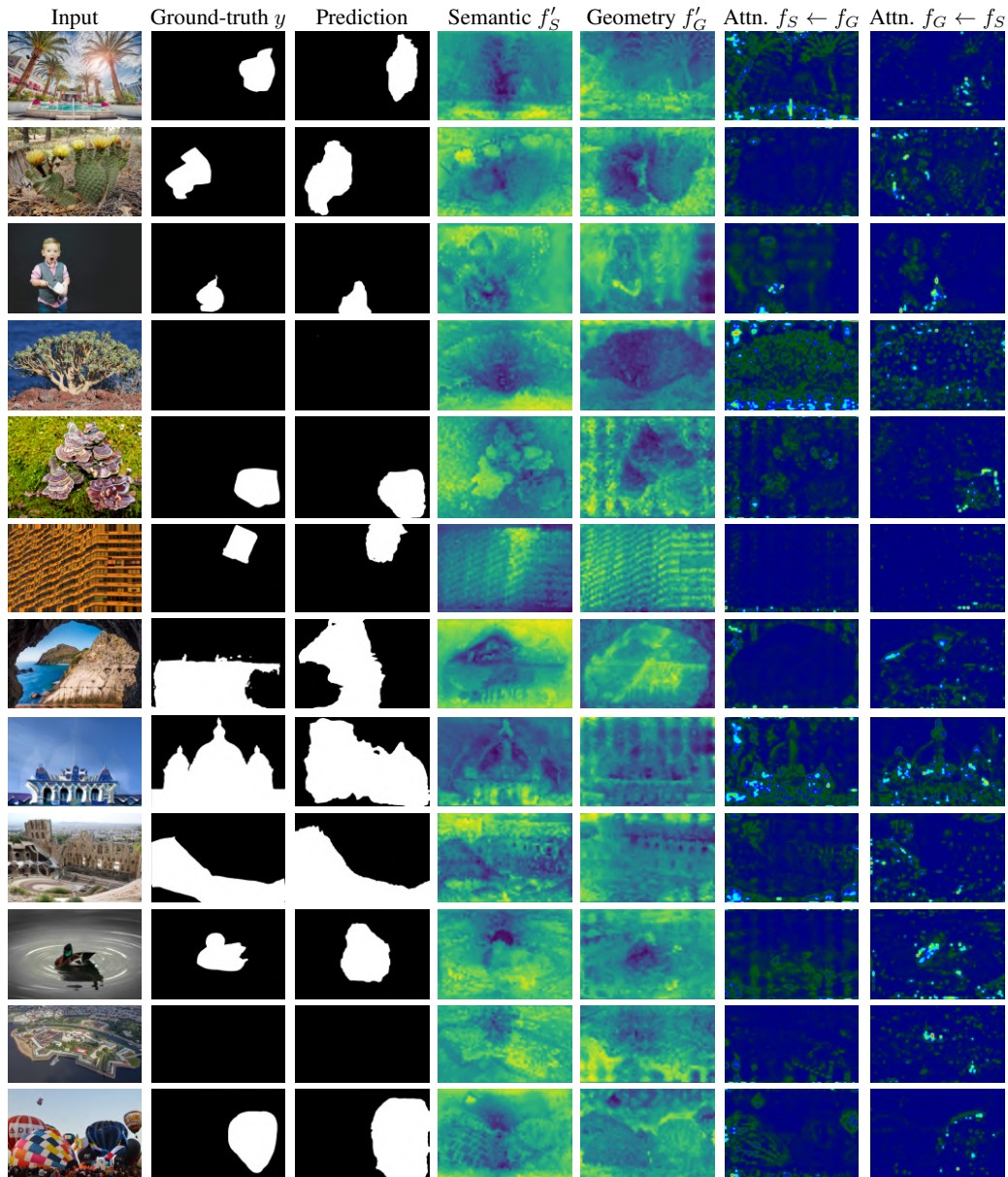

Figure 7: **Features and attention maps.**

Table 9: **Inference time vs. Memory Occupancy vs. Input resolution.**

| Property | Ours | HiFi-Net [33] | TruFor [32] | MIML [63] | AdaIFL [48] | Mesorch [94] | SAFIRE [45] | SIDA [39] |
|---|---|---|---|---|---|---|---|---|
| Inference time [ms] | 228.176 | 53.852 | 79.1626 | 72.920 | 142.43 | 41.730 | 3266.566 | 725.787 |
| Memory occupancy [GiB] | 4.404 | 5.708 | 6.187 | 6.312 | 9.218 | 7.611 | 21.500 | 15.572 |
| Input resolution [px] | $896 \times 896$ | $256 \times 256$ | $512 \times 512$ | $768 \times 768$ | $1024 \times 1024$ | $512 \times 512$ | $1024 \times 1024$ | $1024 \times 1024$ |

1. **Performance gap**: despite outperforming existing methods, our approach still exhibits a non-trivial error rate, particularly in complex forgery scenarios (e.g., unseen inpainters). This suggests significant room for improvement in generalisation and robustness;

2. **Conservative localisation bias**: RADAR tends to prioritise precision over recall in localisation, resulting in fewer false positives at the cost of missed detections. While this aligns with forensic applications where false accusations carry high stakes, it may limit sensitivity in scenarios requiring exhaustive forgery identification;

Table 10: **RADAR's performance in low-resource environments.**

| Setup | ID | | | | LoRA | | | | OOD | | | |
|---|---|---|---|---|---|---|---|---|---|---|---|---|
| | AUC↑ | Acc↑ | F1↑ | IoU↑ | AUC↑ | Acc↑ | F1↑ | IoU↑ | AUC↑ | Acc↑ | F1↑ | IoU↑ |
| ViT Small | **0.966** | 0.892 | **0.800** | 0.516 | 0.672 | 0.549 | 0.756 | 0.387 | 0.731 | 0.645 | **0.787** | 0.418 |
| ViT Base (**Ours**) | 0.921 | **0.893** | 0.790 | **0.547** | **0.771** | **0.678** | **0.789** | **0.474** | **0.746** | **0.689** | 0.773 | **0.422** |

Table 11: **Full quantitative comparison on BBC-PAIR-ID.**

| Method | Stable Diffusion 1.4 Det. | | Loc. | | Stable Diffusion 1.5 Det. | | Loc. | | Stable Diffusion 2.1 Det. | | Loc. | | Stable Diffusion 3 Det. | | Loc. | | Stable Diffusion 3.5 Det. | | Loc. | |
|---|---|---|---|---|---|---|---|---|---|---|---|---|---|---|---|---|---|---|---|---|
| | AUC↑ | Acc↑ | F1↑ | IoU↑ | AUC↑ | Acc↑ | F1↑ | IoU↑ | AUC↑ | Acc↑ | F1↑ | IoU↑ | AUC↑ | Acc↑ | F1↑ | IoU↑ | AUC↑ | Acc↑ | F1↑ | IoU↑ |
| HiFi-Net [33] | 0.556 | 0.500 | 0.757 | 0.378 | 0.636 | 0.520 | 0.742 | 0.372 | 0.575 | 0.505 | 0.754 | 0.377 | 0.596 | 0.505 | 0.759 | 0.382 | 0.653 | 0.510 | 0.723 | 0.362 |
| TruFor [32] | 0.657 | 0.565 | 0.648 | 0.361 | 0.593 | 0.554 | 0.691 | 0.392 | 0.576 | 0.544 | 0.707 | 0.393 | 0.543 | 0.559 | 0.719 | 0.417 | 0.519 | 0.525 | 0.778 | 0.471 |
| MIML [63] | 0.511 | 0.495 | 0.221 | 0.114 | 0.530 | 0.485 | 0.211 | 0.108 | 0.494 | 0.505 | 0.222 | 0.114 | 0.449 | 0.485 | 0.214 | 0.110 | 0.477 | 0.495 | 0.210 | 0.107 |
| AdaIFL [48] | 0.561 | 0.520 | 0.764 | 0.383 | 0.567 | 0.505 | 0.762 | 0.385 | 0.509 | 0.505 | 0.758 | 0.381 | 0.595 | 0.530 | 0.748 | 0.377 | 0.514 | 0.490 | 0.764 | 0.383 |
| Mesorch [94] | 0.667 | 0.610 | 0.698 | 0.367 | 0.568 | 0.544 | 0.720 | 0.379 | 0.726 | 0.650 | 0.744 | 0.400 | 0.545 | 0.545 | 0.654 | 0.391 | 0.536 | 0.550 | 0.776 | 0.424 |
| SAFIRE [45] | 0.598 | 0.530 | 0.699 | 0.375 | 0.597 | 0.525 | 0.664 | 0.381 | 0.605 | 0.540 | 0.672 | 0.363 | 0.617 | 0.535 | 0.648 | 0.358 | 0.625 | 0.535 | 0.649 | 0.370 |
| SIDA [39] | 0.590 | 0.590 | 0.694 | 0.460 | 0.610 | 0.610 | 0.800 | 0.565 | 0.630 | 0.630 | 0.749 | 0.515 | 0.615 | 0.615 | 0.826 | 0.573 | 0.540 | 0.540 | 0.690 | 0.460 |
| RADAR | 0.938 | 0.935 | 0.813 | 0.564 | 0.950 | 0.935 | 0.823 | 0.580 | 0.939 | 0.935 | 0.824 | 0.580 | 0.941 | 0.931 | 0.825 | 0.583 | 0.944 | 0.929 | 0.831 | 0.598 |

| Method | Stable Diffusion XL Det. | | Loc. | | Kandinsky 2.2 Det. | | Loc. | | Kandinsky 3.1 Det. | | Loc. | | FLUX.1-schnell Det. | | Loc. | | FLUX.1-dev Det. | | Loc. | |
|---|---|---|---|---|---|---|---|---|---|---|---|---|---|---|---|---|---|---|---|---|
| | AUC↑ | Acc↑ | F1↑ | IoU↑ | AUC↑ | Acc↑ | F1↑ | IoU↑ | AUC↑ | Acc↑ | F1↑ | IoU↑ | AUC↑ | Acc↑ | F1↑ | IoU↑ | AUC↑ | Acc↑ | F1↑ | IoU↑ |
| HiFi-Net [33] | 0.668 | 0.510 | 0.726 | 0.364 | 0.700 | 0.510 | 0.765 | 0.389 | 0.624 | 0.505 | 0.746 | 0.373 | 0.630 | 0.505 | 0.770 | 0.385 | 0.607 | 0.515 | 0.757 | 0.379 |
| TruFor [32] | 0.525 | 0.520 | 0.779 | 0.475 | 0.499 | 0.534 | 0.780 | 0.473 | 0.518 | 0.530 | 0.757 | 0.449 | 0.494 | 0.510 | 0.779 | 0.472 | 0.558 | 0.534 | 0.753 | 0.432 |
| MIML [63] | 0.469 | 0.485 | 0.207 | 0.106 | 0.469 | 0.485 | 0.215 | 0.110 | 0.390 | 0.435 | 0.219 | 0.111 | 0.472 | 0.490 | 0.197 | 0.102 | 0.419 | 0.465 | 0.221 | 0.113 |
| AdaIFL [48] | 0.520 | 0.475 | 0.766 | 0.384 | 0.506 | 0.470 | 0.766 | 0.384 | 0.523 | 0.475 | 0.765 | 0.383 | 0.505 | 0.505 | 0.766 | 0.383 | 0.531 | 0.505 | 0.760 | 0.383 |
| Mesorch [94] | 0.525 | 0.540 | 0.780 | 0.420 | 0.537 | 0.545 | 0.759 | 0.405 | 0.550 | 0.510 | 0.773 | 0.402 | 0.615 | 0.585 | 0.763 | 0.429 | 0.537 | 0.520 | 0.770 | 0.405 |
| SAFIRE [45] | 0.565 | 0.540 | 0.634 | 0.343 | 0.572 | 0.520 | 0.641 | 0.378 | 0.559 | 0.515 | 0.706 | 0.404 | 0.591 | 0.525 | 0.708 | 0.412 | 0.607 | 0.540 | 0.662 | 0.364 |
| SIDA [39] | 0.550 | 0.550 | 0.775 | 0.564 | 0.735 | 0.735 | 0.772 | 0.558 | 0.565 | 0.565 | 0.783 | 0.523 | 0.555 | 0.555 | 0.820 | 0.674 | 0.590 | 0.590 | 0.728 | 0.465 |
| RADAR | 0.946 | 0.929 | 0.831 | 0.598 | 0.947 | 0.930 | 0.840 | 0.616 | 0.947 | 0.930 | 0.841 | 0.617 | 0.949 | 0.929 | 0.847 | 0.630 | 0.949 | 0.929 | 0.847 | 0.629 |

3. **Explainability**: although we analyse cross-attention maps and activation patterns to interpret model decisions, the internal reasoning remains opaque. A more intuitive, human-aligned explanation framework would improve trust and usability in critical domains like journalism or law;

4. **Evaluation metrics**: the IFDL field lacks standardised benchmarks and metrics, particularly for localisation granularity (e.g., pixel-wise vs. region-wise accuracy) and real-world robustness. Community-wide efforts to establish unified evaluation protocols would better guide future research.

In future work, we aim to address these limitations by (1) exploring even more expressive architectures to reduce error rates, (2) designing adaptive loss functions to balance precision and recall, (3) integrating post-hoc explainability methods (e.g., concept activation vectors), and (4) collaborating with the IFDL community to define standardized evaluation protocols.

## D.2 Broader impact

The development of robust image forgery detection and localisation techniques has significant societal implications. As digital images play a crucial role in journalism, legal evidence, social media, and historical documentation, the ability to reliably detect manipulated content helps combat misinformation, deepfakes, and fraudulent media. This research contributes to the broader effort of ensuring digital media integrity, fostering trust in visual information, and supporting forensic investigations.

However, the deployment of such technologies also raises ethical concerns. Malicious actors could potentially misuse forgery detection models to refine adversarial attacks, making manipulated images even harder to detect. Additionally, false positives in forgery detection could lead to unjust accusations, particularly in sensitive domains like law enforcement or public discourse. To mitigate these risks, it is essential to ensure transparency in model limitations, promote responsible use, and integrate human oversight in critical decision-making processes.

Future advancements in this field should prioritise fairness, avoiding biases that may disproportionately affect certain demographics or media sources. Collaborative efforts between researchers, policymakers, and industry stakeholders will be necessary to establish ethical guidelines and regulatory frameworks for the responsible deployment of image forensics tools.

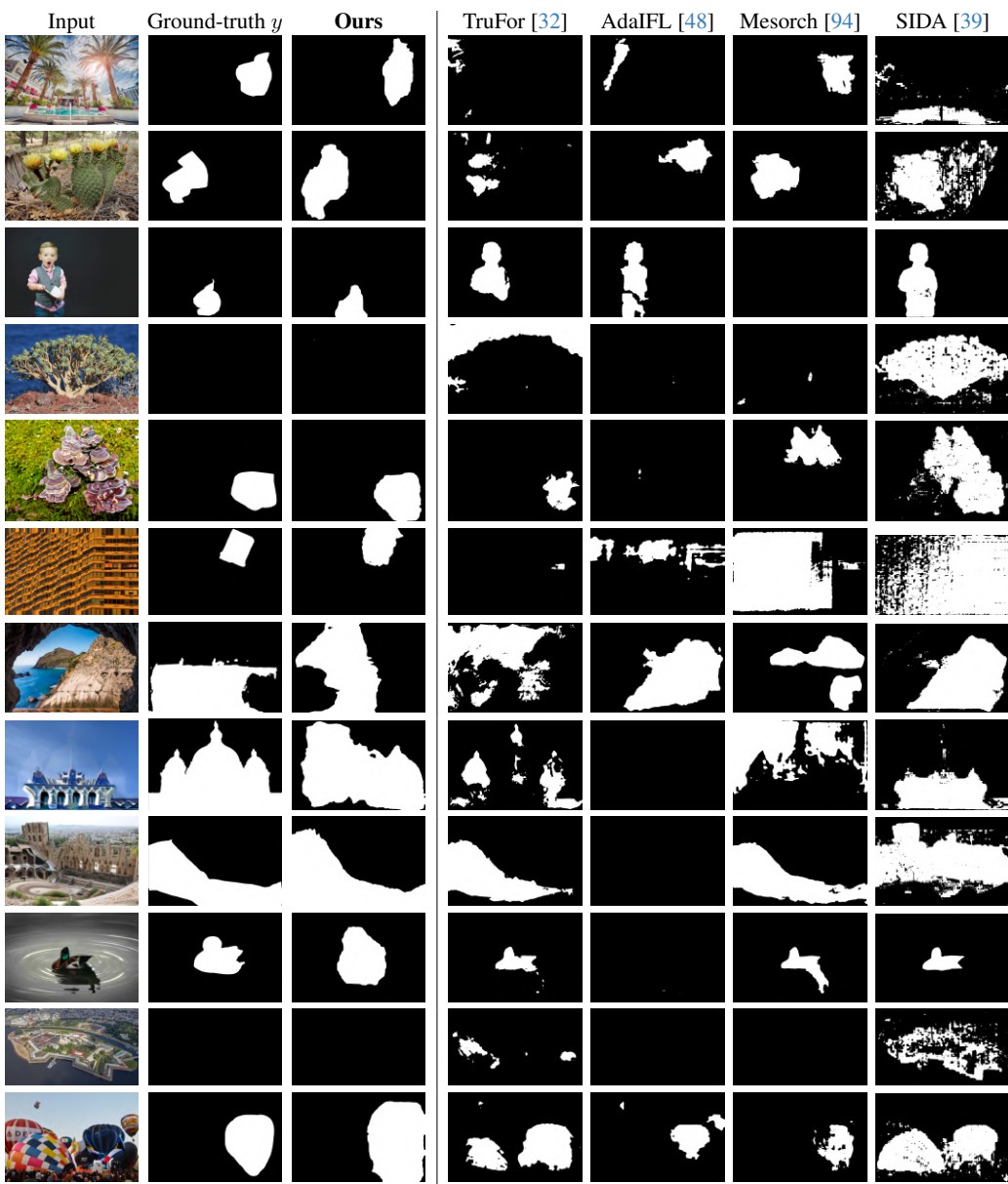

Figure 8: **Qualitative results on BBC-PAIR-OOD (closed-source inpainters).**

By advancing the state of the art in forgery detection, this work not only enhances media authenticity but also encourages further research into trustworthy multimedia systems, ultimately benefiting society at large.

### D.3 Safeguards

Due to the nature of the topic, the proposed method may be vulnerable to malicious exploitation, such as adversarial attacks designed to produce even more convincing image tampering that can evade IFDL techniques. To mitigate this risk, users will be required to follow specific usage guidelines when accessing both the training data and the trained models.

Furthermore, all images in the dataset will include C2PA metadata to facilitate usage tracking. When a user accesses the dataset, C2PA metadata will automatically record the user's name and affiliated institution in the manifest, thereby promoting transparency and fair use. If the C2PA metadata is

Table 12: **Full quantitative comparison on BBC-PAIR-LoRA.**

| Method | dAIversity Detailer 1.5 Det. AUC↑ | Acc↑ | Loc. F1↑ | IoU↑ | dAIversity SD3.5-LoRA Det. AUC↑ | Acc↑ | Loc. F1↑ | IoU↑ | Flux-Detailer-LoRA Det. AUC↑ | Acc↑ | Loc. F1↑ | IoU↑ | Dreamshaper SDXL-1-0 Det. AUC↑ | Acc↑ | Loc. F1↑ | IoU↑ | Juggernaut-XL-v6 Det. AUC↑ | Acc↑ | Loc. F1↑ | IoU↑ |
|---|---|---|---|---|---|---|---|---|---|---|---|---|---|---|---|---|---|---|---|---|
| HiFi-Net [33] | 0.622 | 0.500 | 0.739 | 0.369 | 0.756 | 0.510 | 0.708 | 0.355 | 0.659 | 0.500 | 0.772 | 0.365 | 0.603 | 0.500 | 0.736 | 0.368 | 0.614 | 0.500 | 0.731 | 0.366 |
| TruFor [32] | 0.510 | 0.520 | 0.760 | 0.424 | 0.516 | 0.550 | 0.772 | 0.469 | 0.573 | 0.530 | 0.771 | 0.456 | 0.498 | 0.530 | 0.765 | 0.448 | 0.490 | 0.520 | 0.773 | 0.459 |
| MIML [63] | 0.520 | 0.500 | 0.228 | 0.133 | 0.577 | 0.510 | 0.242 | 0.135 | 0.531 | 0.500 | 0.244 | 0.125 | 0.554 | 0.520 | 0.251 | 0.146 | 0.509 | 0.530 | 0.244 | 0.136 |
| AdaIFL [48] | 0.514 | 0.500 | 0.759 | 0.399 | 0.526 | 0.500 | 0.761 | 0.414 | 0.549 | 0.540 | 0.759 | 0.408 | 0.588 | 0.580 | 0.784 | 0.452 | 0.510 | 0.530 | 0.761 | 0.403 |
| Mesorch [94] | 0.594 | 0.530 | 0.763 | 0.406 | 0.536 | 0.510 | 0.753 | 0.389 | 0.549 | 0.490 | 0.753 | 0.399 | 0.627 | 0.560 | 0.758 | 0.402 | 0.542 | 0.520 | 0.757 | 0.393 |
| SAFIRE [45] | 0.500 | 0.490 | 0.676 | 0.399 | 0.528 | 0.490 | 0.676 | 0.399 | 0.528 | 0.540 | 0.681 | 0.396 | 0.572 | 0.530 | 0.652 | 0.384 | 0.492 | 0.490 | 0.709 | 0.424 |
| SIDA [39] | 0.520 | 0.520 | 0.744 | 0.539 | 0.540 | 0.540 | 0.806 | 0.660 | 0.510 | 0.510 | 0.219 | 0.107 | 0.570 | 0.570 | 0.768 | 0.538 | 0.600 | 0.600 | 0.852 | 0.636 |
| RADAR | 0.944 | 0.840 | 0.821 | 0.547 | 0.930 | 0.820 | 0.820 | 0.552 | 0.891 | 0.770 | 0.808 | 0.523 | 0.892 | 0.760 | 0.807 | 0.513 | 0.895 | 0.760 | 0.807 | 0.513 |

| Method | Perfection 1.5 Det. AUC↑ | Acc↑ | Loc. F1↑ | IoU↑ | Perfection Flux Det. AUC↑ | Acc↑ | Loc. F1↑ | IoU↑ | flux-RealismLora Det. AUC↑ | Acc↑ | Loc. F1↑ | IoU↑ | ReaPhoLoRA 1.5 Det. AUC↑ | Acc↑ | Loc. F1↑ | IoU↑ | Yamer's Realsim-v2 XL Det. AUC↑ | Acc↑ | Loc. F1↑ | IoU↑ |
|---|---|---|---|---|---|---|---|---|---|---|---|---|---|---|---|---|---|---|---|---|
| HiFi-Net [33] | 0.642 | 0.500 | 0.732 | 0.366 | 0.666 | 0.510 | 0.711 | 0.361 | 0.674 | 0.510 | 0.720 | 0.362 | 0.664 | 0.500 | 0.731 | 0.366 | 0.602 | 0.500 | 0.741 | 0.370 |
| TruFor [32] | 0.502 | 0.530 | 0.746 | 0.407 | 0.535 | 0.520 | 0.762 | 0.433 | 0.560 | 0.540 | 0.765 | 0.442 | 0.499 | 0.530 | 0.756 | 0.426 | 0.505 | 0.520 | 0.776 | 0.470 |
| MIML [63] | 0.473 | 0.480 | 0.224 | 0.120 | 0.507 | 0.490 | 0.244 | 0.124 | 0.507 | 0.500 | 0.244 | 0.124 | 0.489 | 0.459 | 0.234 | 0.134 | 0.503 | 0.510 | 0.241 | 0.137 |
| AdaIFL [48] | 0.498 | 0.500 | 0.755 | 0.391 | 0.570 | 0.560 | 0.756 | 0.405 | 0.577 | 0.570 | 0.760 | 0.406 | 0.532 | 0.490 | 0.758 | 0.408 | 0.538 | 0.550 | 0.777 | 0.434 |
| Mesorch [94] | 0.554 | 0.520 | 0.764 | 0.399 | 0.549 | 0.540 | 0.753 | 0.400 | 0.563 | 0.530 | 0.755 | 0.393 | 0.556 | 0.530 | 0.760 | 0.400 | 0.533 | 0.460 | 0.761 | 0.399 |
| SAFIRE [45] | 0.510 | 0.530 | 0.641 | 0.363 | 0.469 | 0.510 | 0.657 | 0.374 | 0.502 | 0.530 | 0.679 | 0.416 | 0.531 | 0.520 | 0.663 | 0.378 | 0.513 | 0.520 | 0.700 | 0.417 |
| SIDA [39] | 0.550 | 0.550 | 0.696 | 0.477 | 0.530 | 0.530 | 0.957 | 0.822 | 0.520 | 0.520 | 0.210 | 0.111 | 0.610 | 0.610 | 0.663 | 0.509 | 0.590 | 0.590 | 0.888 | 0.685 |
| RADAR | 0.903 | 0.771 | 0.808 | 0.516 | 0.892 | 0.755 | 0.806 | 0.511 | 0.882 | 0.743 | 0.803 | 0.504 | 0.892 | 0.758 | 0.810 | 0.517 | 0.893 | 0.756 | 0.809 | 0.515 |

Table 13: **Full quantitative comparison on BBC-PAIR-Commercial.**

| Method | ClipDrop Det. AUC↑ | Acc↑ | Loc. F1↑ | IoU↑ | Dall-E 2 Det. AUC↑ | Acc↑ | Loc. F1↑ | IoU↑ | Adobe Firefly Det. AUC↑ | Acc↑ | Loc. F1↑ | IoU↑ | FLUX.1 Fill [pro] Det. AUC↑ | Acc↑ | Loc. F1↑ | IoU↑ |
|---|---|---|---|---|---|---|---|---|---|---|---|---|---|---|---|---|
| HiFi-Net [33] | 0.344 | 0.500 | 0.815 | 0.413 | 0.582 | 0.513 | 0.675 | 0.399 | 0.531 | 0.513 | 0.750 | 0.405 | 0.372 | 0.500 | 0.773 | 0.413 |
| TruFor [32] | 0.605 | 0.546 | 0.800 | 0.481 | 0.563 | 0.520 | 0.757 | 0.418 | 0.499 | 0.493 | 0.780 | 0.419 | 0.442 | 0.500 | 0.762 | 0.419 |
| MIML [63] | 0.704 | 0.500 | 0.178 | 0.096 | 0.571 | 0.493 | 0.184 | 0.097 | 0.528 | 0.500 | 0.186 | 0.098 | 0.561 | 0.500 | 0.184 | 0.097 |
| AdaIFL [48] | 0.518 | 0.513 | 0.804 | 0.421 | 0.522 | 0.506 | 0.805 | 0.422 | 0.500 | 0.513 | 0.809 | 0.424 | 0.468 | 0.460 | 0.810 | 0.422 |
| Mesorch [94] | 0.572 | 0.526 | 0.818 | 0.433 | 0.562 | 0.553 | 0.811 | 0.419 | 0.521 | 0.540 | 0.813 | 0.412 | 0.527 | 0.553 | 0.806 | 0.417 |
| SAFIRE [45] | 0.422 | 0.493 | 0.674 | 0.368 | 0.526 | 0.500 | 0.702 | 0.369 | 0.513 | 0.506 | 0.685 | 0.396 | 0.424 | 0.500 | 0.673 | 0.405 |
| SIDA [39] | 0.493 | 0.493 | 0.696 | 0.411 | 0.486 | 0.486 | 0.695 | 0.396 | 0.546 | 0.546 | 0.703 | 0.409 | 0.513 | 0.513 | 0.709 | 0.412 |
| RADAR | 0.682 | 0.620 | 0.804 | 0.430 | 0.685 | 0.586 | 0.800 | 0.420 | 0.642 | 0.555 | 0.806 | 0.420 | 0.638 | 0.550 | 0.808 | 0.422 |

| Method | Ideogram Det. AUC↑ | Acc↑ | Loc. F1↑ | IoU↑ | LightX Det. AUC↑ | Acc↑ | Loc. F1↑ | IoU↑ | Phot.AI Det. AUC↑ | Acc↑ | Loc. F1↑ | IoU↑ | YouCam Det. AUC↑ | Acc↑ | Loc. F1↑ | IoU↑ |
|---|---|---|---|---|---|---|---|---|---|---|---|---|---|---|---|---|
| HiFi-Net [33] | 0.374 | 0.500 | 0.764 | 0.413 | 0.460 | 0.500 | 0.806 | 0.413 | 0.320 | 0.500 | 0.799 | 0.414 | 0.388 | 0.500 | 0.815 | 0.411 |
| TruFor [32] | 0.476 | 0.513 | 0.805 | 0.419 | 0.663 | 0.580 | 0.761 | 0.470 | 0.507 | 0.533 | 0.825 | 0.450 | 0.462 | 0.493 | 0.746 | 0.394 |
| MIML [63] | 0.501 | 0.493 | 0.188 | 0.099 | 0.823 | 0.506 | 0.165 | 0.091 | 0.634 | 0.500 | 0.166 | 0.088 | 0.694 | 0.500 | 0.194 | 0.101 |
| AdaIFL [48] | 0.483 | 0.493 | 0.804 | 0.413 | 0.580 | 0.573 | 0.787 | 0.413 | 0.491 | 0.506 | 0.809 | 0.422 | 0.452 | 0.460 | 0.803 | 0.412 |
| Mesorch [94] | 0.526 | 0.526 | 0.807 | 0.406 | 0.782 | 0.706 | 0.809 | 0.495 | 0.593 | 0.573 | 0.808 | 0.422 | 0.525 | 0.526 | 0.801 | 0.403 |
| SAFIRE [45] | 0.447 | 0.486 | 0.690 | 0.371 | 0.584 | 0.506 | 0.647 | 0.395 | 0.468 | 0.500 | 0.693 | 0.428 | 0.537 | 0.500 | 0.722 | 0.428 |
| SIDA [39] | 0.493 | 0.493 | 0.714 | 0.418 | 0.553 | 0.553 | 0.695 | 0.404 | 0.546 | 0.546 | 0.711 | 0.415 | 0.486 | 0.486 | 0.685 | 0.374 |
| RADAR | 0.623 | 0.538 | 0.810 | 0.421 | 0.638 | 0.554 | 0.805 | 0.421 | 0.642 | 0.556 | 0.805 | 0.421 | 0.639 | 0.553 | 0.804 | 0.420 |

removed, a hash-based verification mechanism will be provided: users can query a list of hashes to determine whether a suspicious image originated from the dataset.

Finally, while efforts have been made to filter sensitive content, there is no absolute guarantee that NSFW (Not Safe For Work) images have not been generated during the inpainting process. To help mitigate this risk, all images have been analysed using the default Stability AI safety checker, both before and after inpainting. Relevant flags are included in the dataset's metadata, allowing users to filter out potentially inappropriate content if desired.

Table 14: **Full quantitative comparison on BBC-PAIR-OOD.** Official results from original paper marked in blue.

| Method | BBC-PAIR-Comm | | | | CocoGlide | | | | SID-Set | | | | SafireMS-Expert++ | | | |
|---|---|---|---|---|---|---|---|---|---|---|---|---|---|---|---|---|
| | Det. | | Loc. | | Det. | | Loc. | | Det. | | Loc. | | Det. | | Loc. | |
| | AUC↑ | Acc↑ | F1↑ | IoU↑ | AUC↑ | Acc↑ | F1↑ | IoU↑ | AUC↑ | Acc↑ | F1↑ | IoU↑ | AUC↑ | Acc↑ | F1↑ | IoU↑ |
| HiFi-Net [33] | 0.421 | 0.503 | 0.775 | 0.410 | 0.632 | 0.534 | 0.519 | 0.343 | 0.518 | 0.503 | 0.843 | 0.463 | 0.726 | 0.558 | 0.620 | 0.313 |
| TruFor [32] | 0.527 | 0.522 | 0.780 | 0.434 | 0.752 | 0.647 | 0.711 | 0.381 | 0.460 | 0.514 | 0.907 | 0.467 | 0.700 | 0.635 | 0.610 | 0.467 |
| | | | | | 0.752 | 0.639 | 0.720 | N.A. | | | | | | | | |
| MIML [63] | 0.627 | 0.499 | 0.181 | 0.096 | 0.763 | 0.664 | 0.276 | 0.147 | 0.280 | 0.303 | 0.077 | 0.040 | 0.845 | 0.552 | 0.214 | 0.111 |
| AdaIFL [48] | 0.502 | 0.503 | 0.804 | 0.419 | 0.555 | 0.535 | 0.701 | 0.376 | 0.534 | 0.514 | 0.909 | 0.466 | 0.832 | 0.665 | 0.749 | 0.527 |
| Mesorch [94] | 0.576 | 0.563 | 0.809 | 0.426 | 0.727 | 0.640 | 0.704 | 0.374 | 0.567 | 0.525 | 0.920 | 0.468 | 0.797 | 0.711 | 0.727 | 0.498 |
| SAFIRE [45] | 0.490 | 0.499 | 0.686 | 0.395 | 0.557 | 0.519 | 0.647 | 0.359 | 0.510 | 0.500 | 0.784 | 0.407 | 0.814 | 0.509 | 0.747 | 0.596 |
| | | | | | N.A. | N.A. | 0.635 | N.A. | | | | | | | | |
| SIDA [39] | 0.515 | 0.515 | 0.701 | 0.405 | 0.625 | 0.625 | 0.650 | 0.376 | 0.915 | 0.915 | 0.842 | 0.444 | 0.662 | 0.662 | 0.692 | 0.476 |
| | | | | | | | | | N.A. | 0.901 | 0.739 | 0.438 | | | | |
| RADAR | 0.649 | 0.564 | 0.805 | 0.422 | 0.707 | 0.654 | 0.670 | 0.381 | 0.932 | 0.796 | 0.861 | 0.453 | 0.932 | 0.821 | 0.802 | 0.545 |

Table 15: **Full quantitative comparison on OpenSDI.**

| Method | Det. | | Loc. | |
|---|---|---|---|---|
| | AUC↑ | Acc↑ | F1↑ | IoU↑ |
| RADAR | **0.684** | **0.564** | 0.865 | **0.614** |
| SIDA | 0.536 | 0.536 | 0.735 | 0.394 |
| SAFIRE | 0.476 | 0.506 | 0.724 | 0.383 |
| Mesorch | 0.549 | 0.526 | 0.826 | 0.434 |
| AdaIFL | 0.490 | 0.481 | **0.889** | 0.540 |
| MIML | 0.590 | 0.529 | 0.091 | 0.046 |
| TruFor | 0.552 | 0.525 | 0.848 | 0.454 |
| HiFi-Net | 0.496 | 0.499 | 0.831 | 0.415 |

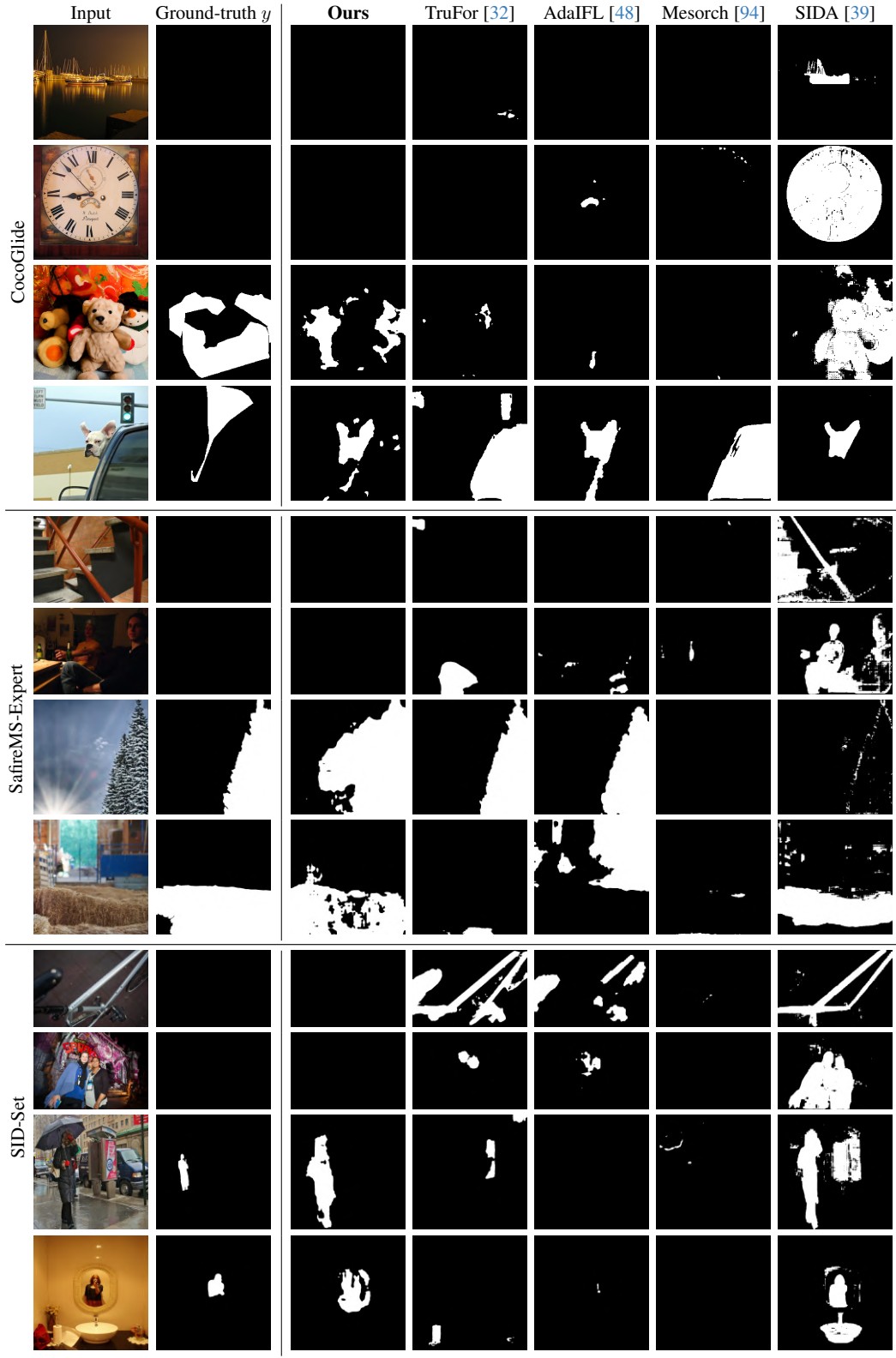

Figure 9: **Qualitative results on other BBC-PAIR-OOD benchmarks.**

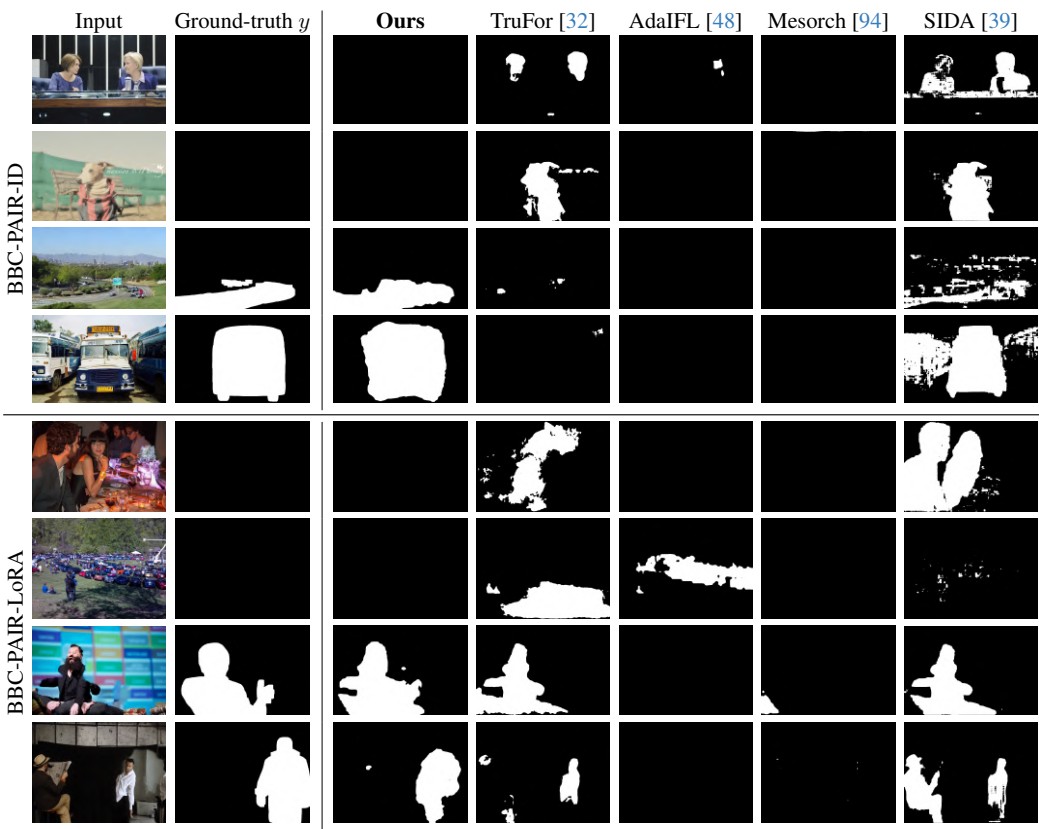

Figure 10: **Qualitative results BBC-PAIR-ID and BBC-PAIR-LoRA benchmarks.**

