# OpenReview forum: "Towards Reliable Identification of Diffusion-based Image Manipulations"
_NeurIPS.cc/2025/Conference — NeurIPS 2025 poster_

### Official Review · Reviewer_pEkt · 2025-07-02

**Clarity:** 3
**Significance:** 3
**Originality:** 3
**Rating:** 5
**Confidence:** 3

**Summary:**

This paper proposes a novel approach, ReliAble iDentification of inpainted AReas (RADAR), and a new comprehensive benchmark, MIBench, for Image Forgery Detection and Localisation (IFDL), particularly targeting diffusion-based image inpainting. The authors also propose a new comprehensive benchmark, MIBench, to evaluate such forgery localization tasks. To construct training and evaluation data, objects are detected using Kosmos-2, masked using Grounded SAM, and then inpainted using various diffusion-based generative models.  RADAR has two feature extractors with different semantic and geometric modalities, a fusion module, and two prediction heads. RADAR is trained using patch-level contrastive learning with tampered patches, affected patches, and original patches. Experiments show RADAR outperforms other methods and has state-of-the-art performance.

**Questions:**

1. Will the author release the model, code, and benchmark?
2. For semantic feature extraction, why do the authors choose DINOv2 instead of CLIP? CLIP could have better semantic features compared with DINOv2 because CLIP is trained with language.

**Ethical Concerns:**

["NO or VERY MINOR ethics concerns only"]

**Final Justification:**

The rebuttal addresses my concerns. I keep my original rating to accept the paper.

**Limitations:**

Yes.

**Quality:**

3

**Strengths And Weaknesses:**

**Strength**
1. This paper introduces a novel approach and benchmark for diffusion-based image manipulations. Releasing the model, code, and benchmark would significantly benefit the research community.
2. This paper is well-written and well-organized.
3. The experiments are overall comprehensive.

**Weaknesses**
1. The claim made in Lines 174–175 of Section 3.3 is not supported by the experiments. While the ablation study indicates that contrastive learning improves performance, the authors do not provide empirical evidence that training the model with data from more inpainters leads to suboptimal performance in fine-grained tasks, such as forgery localisation.
2. [Minor] Double citation in [40] and [41].

---

> ### Author Rebuttal · Authors · 2025-07-30
>
> We thank the reviewer for their positive and encouraging feedback. We are glad that the **novelty of our approach** and the proposed **benchmark for diffusion-based image manipulations** were recognised as **key contributions**. We appreciate the **acknowledgement of the clarity and organisation of the paper**, as well as the **comprehensiveness of the experimental evaluation**. We also value the reviewer's note on the potential impact of releasing the model, code, and benchmark, which we are committed to sharing with the research community.
>
> Regarding the weaknesses, we respond as follows:
>
> _**(1) Claim.**_ We thank the reviewer for pointing this out. Our intention in LL174–175 of Section 3.3 was to highlight that, in RADAR’s case, removing the contrastive loss or using only a single encoder (as shown in Table 2) leads to reduced detection and localisation performance, despite training with a large number of inpainters. In contrast, method [2], which is trained only for detection, benefits from increased diversity of generators. We agree that the original phrasing may overgeneralise and could be misleading. We will revise this claim in the final version to better reflect our specific findings and avoid confusion;
>
> _**(2) Typo.**_ We thank the reviewer for pointing out this typo, we will amend this in the final version.
>
> Regarding the reviewer’s questions, we reply as follows:
>
> _**(1) Release.**_ We confirm that **we will release the model's weights, code for training and inference, and benchmark** upon publication, as stated in line 15 of the main manuscript. We appreciate the reviewer’s interest in this aspect, and we agree that making these resources publicly available is extremely important and will significantly benefit the research community;
>
> _**(2) Why not CLIP?**_. We thank the reviewer for this insightful question. We chose DINO-v2 primarily due to architectural and resolution compatibility with DepthAnything-v2, which facilitates a more intuitive fusion of features in our cross-attention mechanism, and favourable performance compared to CLIP on self-supervised learning benchmarks requiring semantic understanding [1].
> That said, we agree that CLIP’s language supervision may offer rich semantics as well. Indeed, **we conducted an additional experiment** using CLIP (ViT-B/16) instead of DINO-v2, and found that while performance remained reasonable (as shown in the table below), it was slightly lower compared to DINO-v2 in our setup. Importantly, the performance remained competitive with respect to the inference with DepthAnything-v2 only (see Table 2), proving the effectiveness of our novel fusion strategy. Nevertheless, CLIP remains a viable alternative, and we see potential in future exploration of multi-modal supervision within our framework.
>
>
> |    MIBench Split   |     ID    |           |           |           |    LoRA   |           |           |           |    OOD    |           |           |           |
> |:------------------:|:---------:|:---------:|:---------:|:---------:|:---------:|:---------:|:---------:|:---------:|:---------:|:---------:|:---------:|:---------:|
> |         Task       |    Det.   |           |    Loc.   |           |    Det.   |           |    Loc.   |           |    Det.   |           |    Loc.   |           |
> |        Metric      |   AUROC   |    Acc    |     F1    |    IoU    |   AUROC   |    Acc    |     F1    |    IoU    |   AUROC   |    Acc    |     F1    |    IoU    |
> |      CLIP+DA-v2    |   0,901   |   0,844   | **0,802** |   0,503   |   0,691   |   0,596   |   0,758   |   0,402   |   0,706   |   0,627   | **0,799** |   0,412   |
> |DINO-v2+DA-v2 (Ours)| **0,921** | **0,893** |   0,790   | **0,547** | **0,771** | **0,678** | **0,789** | **0,474** | **0,746** | **0,689** |   0,773   | **0,422** |
>
>
> _**References:**_
> * [1] DINOv2: Learning Robust Visual Features without Supervision, TMLR 2024;
> * [2] Community forensics: Using thousands of generators to train fake image detectors, CVPR 2025.

---

> > ### Comment · Reviewer_pEkt · 2025-08-08
> >
> > The rebuttal addresses my concerns. I keep my original rating to accept the paper.

---

> > > ### Author Response · Authors · 2025-08-08
> > >
> > > Dear Reviewer,
> > >
> > > thank you very much for taking the time to review our rebuttal. We are glad that the additional results addressed your concerns.
> > >
> > > Best regards,
> > >
> > > The Authors

---

### Official Review · Reviewer_qaXm · 2025-07-02

**Clarity:** 2
**Significance:** 2
**Originality:** 2
**Rating:** 3
**Confidence:** 4

**Summary:**

This paper proposes RADAR, a method for detecting and localizing image edits generated by both seen and unseen diffusion models. To support realistic evaluation, the authors introduce MIBench, a benchmark consisting of images manipulated by 28 different diffusion models.

**Questions:**

1. The method proposed by the authors lacks a clear motivation, as well as theoretical analysis or justification to support its design.

2. The pipeline appears straightforward and primarily built from existing components. The paper lacks innovation. The authors do not explain why this particular architectural combination is effective. Why does RADAR outperform other existing methods? The paper lacks theoretical explanations for this performance gap.

3. Could there be potential unfairness in the selection of baseline methods? The authors have not explained how the baseline methods (e.g., TruFor, MIML) were implemented. Were they trained and tested on the same datasets? Ensuring fairness in comparison is crucial. The authors should provide detailed descriptions of the implementation details, experimental setup, and any modifications made to the training pipeline of these baseline methods.

4. The generalization capability of RADAR may largely depend on the pretraining of DINOv2 and Depth Anything V2 encoders, rather than the method itself. Have the authors tested whether RADAR performs equally well with alternative encoders, such as replacing DINOv2 with CLIP? This would help verify whether the method is fundamentally effective or heavily dependent on the choice of backbone.

5. The generalization evaluation is only performed on commercial models. However, it is also important to evaluate generalization across open-source models. For instance, the authors could train on a subset of open-source inpainters and test on other unseen open-source models. Furthermore, the testing images are generated using the same pipeline as the training images, with only the model varied. This may result in biased or unfair evaluation. It is recommended that the authors adopt alternative tampering strategies for generating test samples, for example, manual edits or randomized transformations, to better assess robustness.

**Ethical Concerns:**

["NO or VERY MINOR ethics concerns only"]

**Final Justification:**

The authors' responses addressed some of my concerns; however, the technical novelty of the paper remains somewhat limited. As a result, I did not significantly raise the rating.

**Limitations:**

Please see the questions.

**Paper Formatting Concerns:**

There are no major formatting issues in this paper.

**Quality:**

2

**Strengths And Weaknesses:**

Strengths:
The paper addresses an important and timely problem: reliable image forgery detection and localization in the era of high-fidelity diffusion models. It introduces a benchmark (MIBench) covering a set of diffusion-based manipulations.

Weaknesses:
1. The proposed method and its effectiveness lack theoretical explanations.

2. The method lacks a clear motivation for its design choices.

3. The experimental validation is limited to model replacement. Cross-dataset generalization is not evaluated. Moreover, the results for generalization across open-source models are missing.

---

> ### Author Rebuttal · Authors · 2025-07-30
>
> We thank the reviewer for their feedback. We are glad that the **relevance of the problem** we address was recognised, as well as the appreciation of the **introduction of MIBench as a benchmark covering a wide range of diffusion-based manipulations**.
>
> We believe there may be misunderstandings regarding the motivations behind RADAR and our experimental design. To solve them, we address the reviewer's questions below. We hope our answers below will resolve the doubts of the reviewer.
>
> _**(1a) Motivation.**_ We respectfully but strongly disagree. **The motivation for RADAR design choices is inherent to the challenges posed by diffusion-based forgery detection**. As we state in LL44-58, to contrast the ever-growing number of diffusion models available for tampering, we need to both learn effectively from multiple models and promote generalisation to unseen out-of-distribution ones. We mitigate both problems by proposing our three-way contrastive loss (aiding representation learning from various inpainter models) and multimodal feature extraction (promoting generalisation with multiple foundation models).
>
> _**(1b-2a) Theoretical analysis and justification.**_ Although we agree that theoretical justifications would further strengthen our point, we highlight that this is **highly non-standard for the task**, as for many other tasks in computer vision. Indeed, image forgery detection and localisation remains highly experimental and, as a result, none of the competing baselines provide any theoretical justification for their performance. To compensate for this drawback, we proposed a comprehensive analysis section on why our method works (Section 4.3) and extensive empirical results, **as pointed out by all other reviewers**. We hope that the reviewer will reconsider their assessment of our work in light of this.
>
> _**(2b) Novelty.**_ Our core novelty in the methodological design lies in the combination of multiple foundation models for image forgery detection and localisation, suggesting that image forgery localisation **requires more than just high-level semantics understanding and demands sensitivity to geometric inconsistencies as well**. This is motivated by the observation that diffusion-based forgeries can disrupt both modalities in distinct ways (see LL138-145). This novel intuition not only can influence further practices for identifying manipulated images, but is also enabled by non-trivial technical contributions such as our Fusion Block. Our three-way patch-wise contrastive formulation is also technically novel, and it is motivated by the unique effect that diffusion models have on images. Its effectiveness is further justified in Section 4.3 and Table 2. Importantly, please note that pEkt, 9vWL, and 53cS **all highlight the novelty and innovation of our contributions**;
>
> _**(3) Fairness.**_ **We firmly believe that our comparison is fair**. To ensure a transparent evaluation, we extensively reported the performance of RADAR and baselines in the main paper and the appendices on all splits. In our experiments we take special care to ensure fair evaluation by following standard evaluation protocols and common practices. In particular, following [1,2,3,4,5,6,7], we do not retrain the baselines but use the available checkpoints, as we state in Appendix A.1. This practice has been established due to the high computational costs that make retraining baselines unfeasible. Instead, the standard evaluation in image forgery detection and localisation is to train the most robust model possible on a big dataset for a significant time, and compare its performance on out-of-distribution benchmarks to assess generalisation. We did this extensively with MIBench, further distinguishing also near OOD scenarios (obtained with LoRAs) and far OOD ones (composed of commercial inpainters and existing datasets);
>
> _**(4) Replacing DINO-v2 with CLIP.**_ Thanks for the interesting idea. We replaced DINO-v2 with CLIP as suggested, and report the performance below:
>
> |    MIBench Split   |     ID    |           |           |           |    LoRA   |           |           |           |    OOD    |           |           |           |
> |:------------------:|:---------:|:---------:|:---------:|:---------:|:---------:|:---------:|:---------:|:---------:|:---------:|:---------:|:---------:|:---------:|
> |         Task       |    Det.   |           |    Loc.   |           |    Det.   |           |    Loc.   |           |    Det.   |           |    Loc.   |           |
> |        Metric      |   AUROC   |    Acc    |     F1    |    IoU    |   AUROC   |    Acc    |     F1    |    IoU    |   AUROC   |    Acc    |     F1    |    IoU    |
> |      CLIP+DA-v2    |   0,901   |   0,844   | **0,802** |   0,503   |   0,691   |   0,596   |   0,758   |   0,402   |   0,706   |   0,627   | **0,799** |   0,412   |
> |DINO-v2+DA-v2 (Ours)| **0,921** | **0,893** |   0,790   | **0,547** | **0,771** | **0,678** | **0,789** | **0,474** | **0,746** | **0,689** |   0,773   | **0,422** |
>
> These results indicate that our method is fundamentally effective and able to combine features coming from different encoders. Indeed, results with CLIP are still competitive, but we notice that DINOv2 is performing best, consistent with benchmark evaluations in which it is outperforming CLIP in tasks requiring semantic understanding [8].
>
> _**(5) Generalisation evaluation.**_ Please note that we included **both** commercial **and** open source inpainters in the OOD set. Indeed, MIBench-OOD is composed of both commercial models, and existing datasets with other generation pipelines (as stated in LL246-254), avoiding potential biases in the evaluation. The detailed results on the splits of MIBench-OOD (Table 12 in the Appendix) show that RADAR performs the best in both cases.
>
> We also made an additional effort by proposing the experiment in Figure 6 of the Appendix, using a graphic designer to modify images with manual edits. Let us also highlight the experiment we performed in 9vWL's reply, in which we tested RADAR and existing baselines on the very recent OpenSDI [9], still outperforming all. We believe that this proves our strong generalisation capabilities across models and data generation pipelines.
>
> As a further effort to clear the doubt, we have also performed, as suggested, an experiment where RADAR is trained on a subset of open-source inpainters and tested on others. In particular, we trained RADAR on SD1, SD1.5, SD2, SDXL, SD3, with the same protocol stated in LL325-326 and employed for Table 2 of the main paper, and tested SD3.5, Kandisky 2.2, Kandisky 3.1, Flux Schnell, and Flux Dev. We obtained the following results:
>
> |         Task        |  Det. |       |  Loc. |       |
> |:-------------------:|:-----:|:-----:|:-----:|:-----:|
> |        Metric       | AUROC |  Acc  |   F1  |  IoU  |
> |        SD3.5        | 0,891 | 0,718 | 0,790 | 0,450 |
> |        Kandisky 2.2 | 0,885 | 0,705 | 0,788 | 0,446 |
> |     Kandisky 3.1    | 0,867 | 0,687 | 0,786 | 0,439 |
> |     Flux Schnell    | 0,844 | 0,669 | 0,784 | 0,434 |
> |       Flux Dev      | 0,837 | 0,657 | 0,783 | 0,430 |
> |       Average       | 0,865 | 0,687 | 0,786 | 0,440 |
>
> The table above shows **remarkable results compared to the baselines** shown in Table 1 of the main paper, even considering the restricted training setup, further proving the effectiveness of our generalisation-focused design choices.
>
> As a final consideration, let us clarify our design choice: in the design of a method for IFDL, we believe that **if open source diffusion models exist, then they should be used to generate in-distribution training data, since their inclusion in the training set could maximize the detection accuracy in case a malicious user would use them for editing images**. This is coherent with our motivation in LL39-43, which justifies why we did not consider open source models for generalisation evaluation. Contrary, generating data with commercial models leads to considerable additional costs. In any case, we agree that this additional experiment strengthens our point, and we will include the table in the Appendix. Thanks.
>
> _**References:**_
> * [1] Sida: Social media image deepfake detection, localization and explanation with large multimodal model, CVPR 2025;
> * [2] Safire: Segment any forged image region, AAAI 2025;
> * [3] Mesoscopic insights: Orchestrating multi-scale & hybrid architecture for image manipulation localization, AAAI 2025;
> * [4] Adaifl: Adaptive image forgery localization via a dynamic and importance-aware transformer network, ECCV 2025;
> * [5] Towards modern image manipulation localization: A large-scale dataset and novel methods, CVPR 2024;
> * [6] Trufor: Leveraging all-round clues for trustworthy image forgery detection and localization, CVPR 2023;
> * [7] Hierarchical fine-grained image forgery detection and localization, CVPR 2023;
> * [8] DINOv2: Learning Robust Visual Features without Supervision, TMLR 2024;
> * [9] OpenSDI: Spotting Diffusion-Generated Images in the Open World, CVPR 2025.

---

> ### Comment · Reviewer_qaXm · 2025-08-07
>
> Thank you for the authors' responses. Several of my concerns have been addressed. However, the authors assert that the core novelty in the methodological design lies in the combination of multiple foundation models for image forgery detection and localisation. This appears to be more of an innovation in application rather than a methodological breakthrough. Based on the provided responses, I will adjust my rating accordingly.

---

> > ### Author Response · Authors · 2025-08-07
> >
> > Dear Reviewer,
> >
> > we appreciate your updated assessment and are glad that the additional results addressed several of your concerns. Please, let us clarify one last aspect of the importance of our contribution.
> >
> > We believe that the combination of features coming from the two models is novel from both the methodological and application standpoint. On the application, we do agree that it results from the specific nature of the problem that RADAR is tackling.
> > However, the fusion of such features is nontrivial, and it is made possible only by our architectural design based on the Fusion Block, which is strictly methodological.
> >
> > Moreover, let us highlight that the core novelty also includes our three-way patch-wise contrastive formulation, which is specifically designed for this setting, enabling a more discriminative representation that would not be achievable with standard setups.
> >
> > We hope this helps clarify the methodological contribution.
> >
> > Best regards,
> >
> > The authors

---

### Official Review · Reviewer_9vWL · 2025-07-02

**Clarity:** 4
**Significance:** 4
**Originality:** 3
**Rating:** 5
**Confidence:** 5

**Summary:**

The paper introduces a novel method for IDFL (Image Forgery Detection and Localisation). It integrates features extracted from 2 complementary foundational models, one to extract geometric representations and the other to extract semantic information. The method incorporates a Fusion Block that “exploits a symmetric cross-attention mechanism, based on the swapping of keys $K$ in two multi-head attention mechanisms” that process the 2 sets of extracted features, resulting in a set of fused features. The authors introduce a novel contrastive learning framework trained to differentiate between three categories of image patches: original, tampered, and affected, going beyond the traditional binary real vs. fake setup. A new comprehensive benchmark, MIBench, is proposed, consisting of local manipulations using 28 different generators. The benchmark is split into 3 groups: MIBench-ID, MIBench-LoRA (generators from MIBench-ID with LoRA applied) and MIBench-OOD. This separation creates a similar setup with the ID, Near-OOD and Far-OOD setup from distribution shift studies. The proposed method achieves state-of-the-art results on both the newly introduced benchmark and a few existing datasets, which the benchmark incorporates.

**Questions:**

N/a

**Ethical Concerns:**

["NO or VERY MINOR ethics concerns only"]

**Final Justification:**

The new results on OpenSDI strengthen the evaluation of the proposed method across diverse deepfake detection datasets previously proposed by the research community. This directly addresses the primary concern raised in my initial review.

**Limitations:**

yes

**Quality:**

4

**Strengths And Weaknesses:**

Strengths:

$\bullet$ The proposed method is genuinely novel, by introducing a three-way contrastive distinction that extends beyond standard binary approaches and utilising complementary feature extractions in the context of Image Forgery Detection and Localisation. Strong results across various data sources is also a strength.

$\bullet$ The proposed benchmark includes images manipulated by a wide range of inpainters, including commercial ones. This fills a gap in the literature and enables a more comprehensive evaluation for future models, especially for localisation purposes.

$\bullet$ The ablation studies are comprehensive and answer the main questions that pop up regarding RADAR and its performance. These studies also highlight the complementary characteristics of the two types of image encoders employed for the task of IFDL.

$\bullet$ The paper is very well written, structured and easy to follow.

Weaknesses:

$\bullet$ It would be nice if some results on already established datasets and benchmarks could be visible in the main text. In particular (besides the ones you considered in MIBench-OOD, i.e. CocoGlide, SafireMS-Exper and SID-Set), the OpenSDI[1] benchmark could alleviate the concern that “benchmarks rely on a single inpainting model for image generation” (line 104), as it includes data inpainted by 5 different generators and also addresses IDFL setup. Results on this benchmark or similar ones could strengthen RADAR’s performance claims compared to previous methods.

Decision:

I consider this paper to be worthy of acceptance due to the method’s novelty, especially the modelling of the affected patches into the training objective and the curation of a new IFDL benchmark that consists of manipulated images by a variety of generators, including commercial ones.

[1]Wang, Y., et al. (2025). OpenSDI: Spotting Diffusion-Generated Images in the Open World. arXiv preprint arXiv:2503.19653. https://arxiv.org/abs/2503.19653

---

> ### Author Rebuttal · Authors · 2025-07-30
>
> We thank the reviewer for their thorough and encouraging assessment. We are pleased that the aspects of our method, such as the use of **complementary feature extractions** for image forgery detection and localisation and the **three-way contrastive distinction** beyond standard binary setups, were **recognised as strengths and deemed as genuinely novel**. We also appreciate the positive feedback on the **comprehensiveness** of the proposed MIBench benchmark, its **relevance**, and the **clarity and structure** of the manuscript. We are glad that the **ablation studies were found informative**.
>
> We also thank the reviewer for pointing out the OpenSDI benchmark. We only became aware of its existence during its official presentation at CVPR 2025, which occurred after the submission deadline. We agree that including results on OpenSDI helps contextualise RADAR’s performance more broadly. We have since **run RADAR and the considered competitors on OpenSDI**, reporting the results below:
>
> |         |   RADAR   |           |       |           |  SIDA |       |       |       | SAFIRE |       |       |       | Mesorch |       |       |       |
> |:-------:|:---------:|:---------:|:-----:|:---------:|:-----:|:-----:|:-----:|:-----:|:------:|:-----:|:-----:|:-----:|:-------:|:-----:|:-----:|:-----:|
> |   Task  |    Det.   |           |  Loc. |           |  Det. |       |  Loc. |       |  Det.  |       |  Loc. |       |   Det.  |       |  Loc. |       |
> |  Metric |   AUROC   |     Acc   |   F1  |    IoU    | AUROC |   Acc |   F1  |  IoU  |  AUROC |   Acc |   F1  |  IoU  |  AUROC  |   Acc |   F1  |  IoU  |
> | Average | **0,684** | **0,564** | 0,865 | **0,614** | 0,536 | 0,536 | 0,735 | 0,394 |  0,476 | 0,506 | 0,724 | 0,383 |  0,549  | 0,526 | 0,826 | 0,434 |
>
> |         | AdaIFL |       |           |       |  MIML |       |       |       | Trufor |       |       |       | HiFi-Net |       |       |       |
> |:-------:|:------:|:-----:|:---------:|:-----:|:-----:|:-----:|:-----:|:-----:|:------:|:-----:|:-----:|:-----:|:--------:|:-----:|:-----:|:-----:|
> |   Task  |  Det.  |       |    Loc.   |       |  Det. |       |  Loc. |       |  Det.  |       |  Loc. |       |   Det.   |       |  Loc. |       |
> |  Metric |  AUROC |   Acc |     F1    |  IoU  | AUROC |   Acc |   F1  |  IoU  |  AUROC |   Acc |   F1  |  IoU  |   AUROC  |   Acc |   F1  |  IoU  |
> | Average |  0,490 | 0,481 | **0,889** | 0,540 | 0,590 | 0,529 | 0,091 | 0,046 |  0,552 | 0,525 | 0,848 | 0,454 |   0,496  | 0,499 | 0,831 | 0,415 |
>
> We observe that **RADAR attains the best performance in detection** (on both AUROC and Accuracy metrics) **and localisation** (first in IoU, second in F1), proving its **strong generalisation capabilities**.
> We will include these results in the final version of the manuscript.

---

> > ### Comment · Reviewer_9vWL · 2025-08-05
> >
> > I appreciate the authors’ rebuttal and the additional results provided on the OpenSDI dataset, which further demonstrate the strong performance of the proposed method. Based on this, I strengthen my recommendation for acceptance.

---

> > > ### Author Response · Authors · 2025-08-05
> > >
> > > Dear Reviewer,
> > >
> > > thank you very much for taking the time to review our rebuttal and for your follow-up. We truly appreciate your updated assessment and are glad that the additional results addressed your concerns.
> > >
> > > Best regards,
> > >
> > > The Authors

---

### Official Review · Reviewer_53cS · 2025-07-03

**Clarity:** 3
**Significance:** 3
**Originality:** 3
**Rating:** 4
**Confidence:** 4

**Summary:**

This paper proposes RADAR, a framework for detecting and localizing diffusion-based inpainting manipulations. It fuses semantic and geometric features from two pre-trained encoders using a cross-attention module, and introduces a triplet contrastive loss to enhance generalization across various inpainting models. The authors also construct MIBench, a large-scale benchmark with 150K manipulated images from 28 diffusion models. Experiments show that RADAR significantly outperforms existing methods in both detection and localization, and is robust to unseen manipulation tools.

**Questions:**

1. Applicability beyond inpainting manipulations:
Since RADAR is specifically designed for detecting inpainting-based manipulations, could the authors clarify whether the method is applicable or adaptable to other tampering types such as copy-move, splicing, or Style changes? If the method's performance deteriorates in these cases, what extensions or modifications would be needed?
Clarification here could increase the perceived generalizability.

2. Handling of multi-object/multi-region tampering scenarios:
The paper does not explicitly analyze or report performance when multiple objects or regions are tampered with in the same image. Has the method been evaluated under such conditions? If not, what are the anticipated challenges or limitations in such cases?
Evidence of robustness in multi-region scenarios could strengthen the assessment of real-world applicability.

3. Resource cost and deployment feasibility:
The method uses two large frozen encoders and a cross-attention-based fusion module, but the paper lacks information on training/inference time, memory usage, and runtime efficiency. Could the authors provide details on these metrics and discuss possible optimizations for deployment in real-time or resource-constrained environments?

Providing this information would improve the assessment of scalability and practical deployment.

**Ethical Concerns:**

["NO or VERY MINOR ethics concerns only"]

**Final Justification:**

Thank you for the authors' detailed responses. The authors have clarified most of my questions, and I maintain my positive rating.

**Limitations:**

Yes

**Paper Formatting Concerns:**

No major formatting issues are observed.

**Quality:**

3

**Strengths And Weaknesses:**

# Strengths
1. Innovative fusion of semantic and geometric features for enhanced tampering representation.
2. Triplet contrastive loss improves cross-model generalization.
3. A well-designed benchmark (MIBench) with diverse and large-scale manipulated data.
4. Extensive experiments with strong results across multiple scenarios.

# Weaknesses
1. RADAR primarily detects tampering with the inpainting type, and its applicability to other types of editing is not explicitly discussed in the paper. If there are different forms of tampering in practice, the performance of the method may be unknown. This limits the scope of work.
2. Although the authors conducted qualitative experiments to detect tampering, there is still a lack of experimental analysis of multiple tampering scenarios (such as multiple objects being tampered with at the same time). These are real-world but more complex inspection challenges, and it is recommended to expand to these broader application scenarios in the future.
3. Training RADAR involves 150K images, two large frozen encoders, and a heavy fusion block with cross attention layers. The paper does not report training/inference time, making it difficult to assess practical deployability in resource constrained settings.

---

> ### Author Rebuttal · Authors · 2025-07-30
>
> We thank the reviewer for their careful and constructive feedback. We are pleased that the **innovative fusion of semantic and geometric features** for tampering representation, as well as the use of **triplet contrastive loss to improve cross-model generalisation**, were seen as strengths. We also appreciate the recognition of our **efforts in building MIBench**, a diverse and large-scale benchmark, and the value of the **extensive experiments and strong results across multiple scenarios**.
>
> Regarding the reviewer’s questions, we respond as follows:
>
> _**(1) Applicability beyond inpainting.**_ We tested RADAR's performance on datasets including non-inpainting modifications, reporting these results in Table 4 of Section C.3 of the Appendix.
> We observed that localisation performance is not negatively impacted, indeed, **F1 scores are even higher than those obtained on MIBench**. We ascribe this to the less refined copy-paste operations in these datasets, which, compared to inpainting, do not allow for a smooth blending of inserted elements, thereby easing localisation. However, we also report lower detection performance relative to MIBench. We conjecture that this is due to the lack of distinction between genuine, affected, and tampered pixels in non-inpainting modifications, leading to outliers in detection. Nonetheless, since localisation performance remains satisfactory, calibrating the detection score (i.e., threshold tuning of the precision-recall curve on the validation sets) may suffice to improve overall results.
> An alternative way to tackle this would be to train the framework on non-inpainting forgeries as well, although this is out of scope for our problem formulation.
>
> _**(2) Multi-region scenario.**_ We thank the reviewer for the interesting suggestion. Since the Safire-MS Expert subset from MIBench-OOD specifically includes multi-source forgeries, we analysed RADAR's performance on _Single Region_ forgeries (a single connected component in the ground truth) and _Multiple Regions_ (more than one connected component in the ground truth), in the table below:
>
> | Task                |    Det.   |           |    Loc.   |           |
> |:-------------------:|:---------:|:---------:|:---------:|:---------:|
> | Metric              |   AUROC   |    Acc    |     F1    |    IoU    |
> |        _All_        |   0,932   |   0,821   |   0,802   |   0,545   |
> |   _Single Region_   |   0,930   | **0,829** |   0,795   |   0,540   |
> |  _Multiple Regions_ | **0,960** |   0,712   | **0,898** | **0,622** |
>
> We observe that **RADAR's performance remains competitive in both the presence of single regions and multiple regions inpainted**. We hypothesise that this is due to the patch-level features separation promoted by our contrastive formulation.
>
> _**(3) Efficiency and deployment.**_  Thanks for the suggestion. We did not consider low-resource deployment as a core application of RADAR, akin to all considered competitors [1,2,3,4,5,6,7]. However, we believe it is interesting to investigate this scenario, and we conducted the necessary additional tests.
> First, we highlight that we already report inference times and input sizes for RADAR and the considered competitors in Table 7 of Section C.3 of the Appendix.
> Moreover, we measured the memory footprint of all methods, as shown below. RADAR is the method that has the smallest memory footprint.
>
> |   Property   |  Ours | HiFi-Net | Trufor |  MIML | AdaIFL | Mesorch | SAFIRE |  SIDA  |
> |:------------:|:-----:|:--------:|:------:|:-----:|:------:|:-------:|:------:|:------:|
> | Memory [GiB] | 4,404 |   5,708  |  6,187 | 6,312 |  9,218 |  7,611  | 21,500 | 15,572 |
>
> From the results in Table 7 of Section C.3 and the table above, **RADAR performs competitively in low-resource environments**.
>
> As an additional effort to further reduce the computational cost, we propose an additional experiment by substituting the DINO-v2 and DepthAnything-v2 backbones with their smaller variants, for instance, using ViT Small instead of ViT Base. We train this smaller model and compare it using the reduced training protocol described in LL325-326 due to training times of the full model. We report the results of the experiment below:
>
> | MIBench Split         |     ID    |           |           |           |    LoRA   |           |           |           |    OOD    |           |           |           |
> |:---------------------:|:---------:|:---------:|:---------:|:---------:|:---------:|:---------:|:---------:|:---------:|:---------:|:---------:|:---------:|:---------:|
> |          Task         |    Det.   |           |    Loc.   |           |    Det.   |           |    Loc.   |           |    Det.   |           |    Loc.   |           |
> |         Metric        |   AUROC   |    Acc    |     F1    |    IoU    |   AUROC   |    Acc    |     F1    |    IoU    |   AUROC   |    Acc    |     F1    |    IoU    |
> | ViT Small (optimised) | **0,966** |   0,892   | **0,800** |   0,516   |   0,672   |   0,549   |   0,756   |   0,387   |   0,731   |   0,645   | **0,787** |   0,418   |
> | ViT Base (Ours)       |   0,921   | **0,893** |   0,790   | **0,547** | **0,771** | **0,678** | **0,789** | **0,474** | **0,746** | **0,689** |   0,773   | **0,422** |
>
> This substitution represents a good trade-off between performance and efficiency. In particular, this variant yields a runtime memory footprint of 3.106 GiB and an inference time of 99.424 ms, **representing a 30% reduction in memory and a 57% reduction in inference time** compared to the version presented in the main manuscript. Moreover, we are still very competitive in terms of accuracy and localisation compared to baselines in Table 1, even considering the reduced training setup.
>
> Finally, as concerns training times, we did make use of fairly large computational resources due to the complexity of the task. In particular, we trained our framework for 12 days on the hardware specified in Section B.2 of the Appendix.
>
> _**References:**_
> * [1] Sida: Social media image deepfake detection, localization and explanation with large multimodal model, CVPR 2025;
> * [2] Safire: Segment any forged image region, AAAI 2025;
> * [3] Mesoscopic insights: Orchestrating multi-scale & hybrid architecture for image manipulation localization, AAAI 2025;
> * [4] Adaifl: Adaptive image forgery localization via a dynamic and importance-aware transformer network, ECCV 2025;
> * [5] Towards modern image manipulation localization: A large-scale dataset and novel methods, CVPR 2024;
> * [6] Trufor: Leveraging all-round clues for trustworthy image forgery detection and localization, CVPR 2023;
> * [7] Hierarchical fine-grained image forgery detection and localization, CVPR 2023.

---

### Author Response · Authors · 2025-08-05

Dear Reviewers,

thank you for your time and for the helpful reviews.

We would like to kindly follow up, as we have not yet received any replies to our rebuttal. We believe we have addressed all the concerns raised in the initial reviews, and we would greatly appreciate it if you could consider our responses and let us know if any concerns remain.

If our clarifications have resolved your doubts, we hope you will consider it in the final rating.

We remain available for any further questions or discussions.

Regards,

The Authors

---

### Comment · Area_Chair_wZW6 · 2025-08-05
**Please provide your feedback on the authors' rebuttal**

Dear reviewers,

For those who have not responded yet, please take a look at the authors’ rebuttal and update your final scores. Thanks.

Best,

AC

---

### Note · Authors · 2025-08-16

We thank the reviewers and AC for the time and constructive feedback provided throughout the review process. We are pleased that our efforts in the rebuttal and additional experiments, including the new results on the OpenSDI dataset, the ablation with CLIP encoder, and the multiple inpainting regions ablation, have addressed several concerns and strengthened the support from the reviewers.

Before the decision, we would like to remark on the novelty and potential impact of our contributions.

Our combination of features, derived from heterogeneous foundational encoders, is motivated by the novel insight that diffusion-based forgery detection requires a fine-grained understanding of images, attained by employing both semantic and geometric cues. We believe this insight may influence future research. In this context, the fusion of modalities, made possible through our Fusion Block, is a methodological contribution in itself.

Furthermore, our three-way patch-wise contrastive formulation is specifically designed for the diffusion-based forgery localisation scenario. This approach enables a more discriminative representation than standard binary setups and has not been explored before.

Overall, we believe these contributions are both methodologically novel and of broad practical interest, as diffusion-based generative techniques become increasingly widespread. We hope this context will be useful for the AC in the final decision-making process, and we thank all reviewers and the AC once again for their efforts.

---

### Decision · Program_Chairs · 2025-09-17

**Decision:**

Accept (poster)

**Comment:**

This paper proposes RADAR, a novel framework for detecting and localizing diffusion-based inpainting manipulations. The method demonstrates strong performance and makes a valuable contribution to the field. Following the author rebuttal, the majority of the reviewers' concerns have been adequately addressed, and a consensus for acceptance has been reached. Given its sufficient contributions and the positive response to feedback, I recommend acceptance.